# A first chronology for the East GReenland Ice–core Project (EGRIP) over the Holocene and last glacial termination

Seyedhamidreza Mojtabavi[1,2], Frank Wilhelms[1,2], Eliza Cook[3], Siwan Davies[4], Giulia Sinnl[3], Mathias Skov Jensen[3], Dorthe Dahl-Jensen[3,5], Anders Svensson[3], Bo M. Vinther[3], Sepp Kipfstuhl[1,3], Gwydion Jones[4], Nanna B. Karlsson[1,6], Sergio Henrique Faria[7,8,9], Vasileios Gkinis[3], Helle Astrid Kjær[3], Tobias Erhardt[10], Sarah M. P. Berben[11], Kerim H. Nisancioglu[11,12], Iben Koldtoft[3], and Sune Olander Rasmussen[3]

[1]Alfred–Wegener–Institut Helmholtz–Zentrum für Polar– und Meeresforschung, Bremerhaven, Germany
[2]Department of Crystallography, Geoscience Centre, University of Göttingen, 37073 Göttingen, Germany
[3]Physics of Ice, Climate and Earth, Niels Bohr Institute, University of Copenhagen, Denmark
[4]Department of Geography, College of Science, Swansea University, Swansea, Wales, UK
[5]Center for Earth Observation Science, University of Manitoba, Winnipeg, Canada
[6]Geological Survey of Denmark and Greenland, Copenhagen, Denmark
[7]Basque Centre for Climate Change (BC3), 48940 Leioa, Spain
[8]Nagaoka University of Technology, 940-2188 Nagaoka, Japan
[9]IKERBASQUE, Basque Foundation for Science, 48011 Bilbao, Spain
[10]Climate and Environmental Physics, Physics Institute and Oeschger Center for Climate Change Research, University of Bern, Sidlerstrasse 5, 3012 Bern, Switzerland
[11]Department of Earth Science, University of Bergen, Bjerknes Centre for Climate Research, Allégaten 41, 5007, Bergen, Norway
[12]Centre for Earth Evolution and Dynamics, University of Oslo, Oslo, Norway
**Correspondence:** Seyedhamidreza Mojtabavi (Seyedhamidreza.Mojtabavi@awi.de)

**Abstract.**

This paper provides the first chronology for the deep ice core from the East GReenland Ice-core Project (EGRIP) over the Holocene and the late last glacial period. We rely mainly on volcanic events and common peak patterns recorded by dielectric profiling (DEP) and electrical conductivity measurement (ECM) for the synchronization between the EGRIP, NEEM and NGRIP ice cores in Greenland. We transfer the annual-layer-counted Greenland Ice Core Chronology 2005 (GICC05) from the NGRIP core to the EGRIP ice core by means of 381 match points, typically spaced less than 50 years apart. The NEEM ice core has previously been dated in a similar way, and is only included to support the match-point identification. We name our EGRIP time scale GICC05-EGRIP-1. Over the uppermost 1383.84 m, we establish a depth–age relationship dating back to 14,967 a b2k (years before the year 2000 CE). Tephra horizons provide an independent validation of our match points. In addition, we compare the ratio of the annual layer thickness between ice cores in between the match points to assess our results in view of the different ice-flow patterns and accumulation regimes of the different periods and geographical regions. For the next years, this initial timescale will be the basis for climatic reconstructions from EGRIP high-resolution proxy data sets, like e.g. stable water isotopes, chemical impurity or dust records.

# 1 Introduction

The dating of an ice core establishes the depth–age relationship to derive a chronology of past climatic conditions from the measured proxy parameters. The proxy parameters reflect past atmospheric conditions and biogeochemical events along the core. Concerning the ice sheet as a whole, the depth–age relation is needed to map the ice sheet's internal architecture to interpret and understand the climatic evolution and the behaviour of ice streams (MacGregor et al., 2015). This is a particular focus of the East GReenland Ice-core Project (EGRIP). The drill site has been chosen close to the onset of the North East Greenland Ice Stream (NEGIS) (see Fig. 1), which is the largest ice stream of the Greenland ice sheet (Joughin et al., 2010, 2018). A main objective of the EGRIP project is to study the dynamics of the ice flow in the NEGIS ice stream by analysing the ice core's rheology and its relation to the deformation of the ice.

The main objective of this work is to facilitate analysis of the data from the core by transferring the GICC05 timescale, which has already been transferred from NGRIP to the GRIP, GISP2, and NEEM deep ice cores (Seierstad et al., 2014; Rasmussen et al., 2013), to EGRIP by aligning features in the DEP and ECM data sets of the EGRIP, NGRIP and NEEM ice cores. We establish a timescale for the time period of the Holocene and the last glacial termination. We stop specifically at 15 ka b2k, as the density of the match points in the period of Holocene, GS-1 (often called the Younger Dryas) and GI-1 (or the Bølling-Allerød) is much higher than in the glacial (and especially in the Last Glacial Maximum, LGM). We present this timescale to enable climate studies while work on a revised layer-counted timescale is ongoing.

The GICC05modelext timescale was transferred from NGRIP to the North Greenland Eemian (NEEM) ice core by matching 787 match points of mainly volcanic origin identified in the electrical conductivity measurement (ECM) and dielectric profiling (DEP) records and – where available – verified by tephra horizons (Rasmussen et al., 2013). To apply this approach to the EGRIP core, we have profiled the upper 1383.84 m of the EGRIP core using ECM and DEP in the field during the 2017, 2018 and 2019 field seasons. We rely mainly on volcanic events as reflected in the common peak pattern in the DEP and ECM records for the synchronization between the EGRIP ice core and the NGRIP1 and NGRIP2 cores. The NEEM ice core is included in order to support match-point identification, while the GICC05 ages are transferred from NGRIP to EGRIP. Three identified tephra horizons independently verify the correct match of the ice cores.

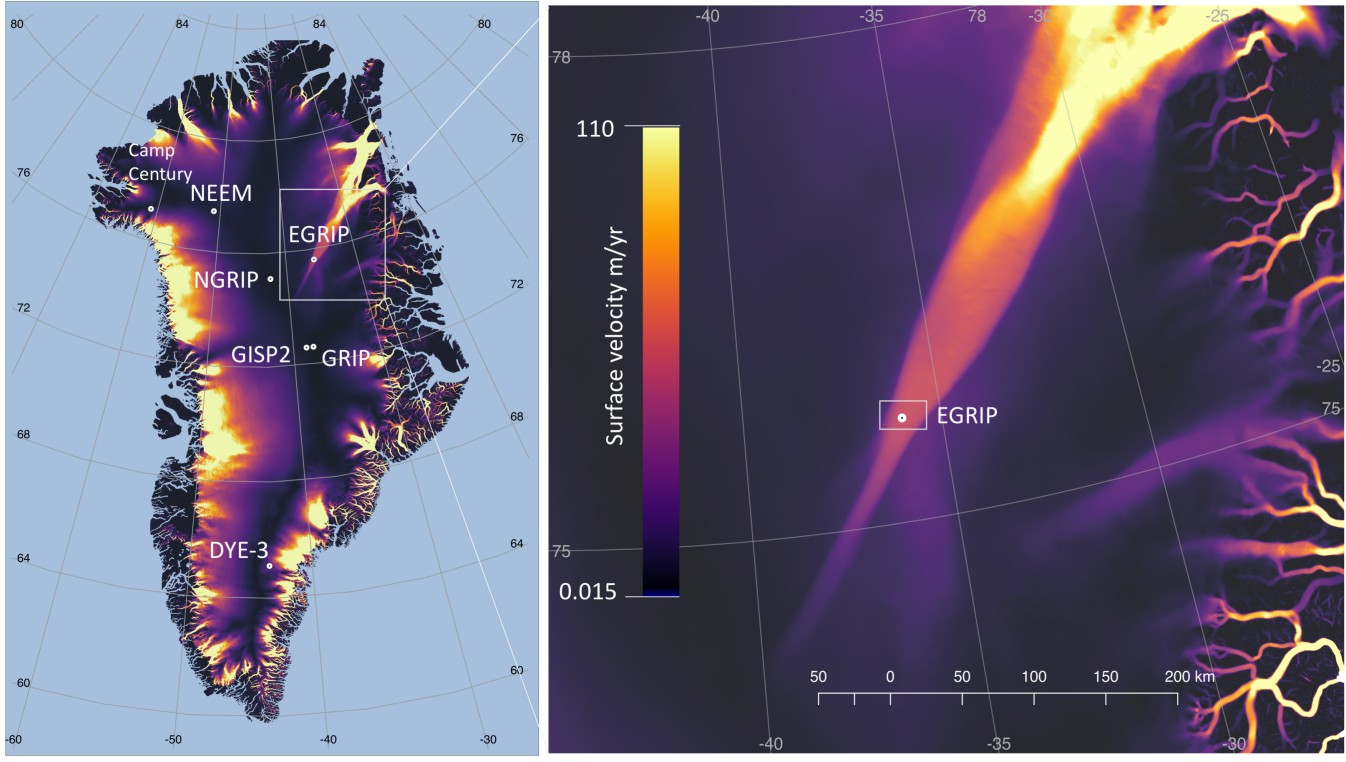

**Figure 1.** Locations of deep ice-core drill sites: EGRIP, NEEM, NGRIP, GRIP, GISP2, DYE–3, and Camp Century in Greenland, and close-up of the EGRIP drill site inside the North East Greenland Ice Stream (NEGIS). Colours show surface flow velocities from satellite data (Joughin et al., 2018).

## 2 Data and methods

### 2.1 GICC05

The annual-layer-counted Greenland Ice Core Chronology 2005 (GICC05) is derived from measurements of stable water isotopes in the DYE–3, GRIP and NGRIP (see Fig. 1) ice cores for the period back to 7.9 ka b2k (Vinther et al., 2006) and high-resolution measurements of chemical impurities, conductivity of the ice, and visual stratigraphy from the GRIP and NGRIP ice cores for the period between 7.9 ka and 14.7 ka b2k (Rasmussen et al., 2006). For the period from 14.7 ka to 42 ka b2k, the dating of the cores is based on annual layer counting in the visual stratigraphy, the electrical conductivity

profiles, and a set of chemical impurities data (Andersen et al., 2006). The timescales are compared to time scales of different other climate archives at suitable tie points, like e.g. marine sediment cores (Svensson et al., 2006). For the NGRIP core, the GICC05 time scale has been extended even further into the glacial, back to 60 ka b2k by annual layer counting (Svensson et al., 2008) and ice-flow modelling (Wolff et al., 2010). For the older parts (Wolff et al., 2010) the NGRIP ss09sea06bm model time scale, shifted to younger ages by 705 years, has been spliced onto the end of the GICC05 timescale, thereby

forming the so-called GICC05modelext chronology. The GICC05modelext was also applied to the central Greenland GRIP and GISP2 cores by more than 900 marker points and verification with 24 tephra horizons (Seierstad et al., 2014). In summary, the GICC05modelext timescale is the consistent reference frame for the entirety of Greenland deep cores.

Previous studies assessed the differences between independent timescales of Holocene paleoclimate records. Adolphi and Muscheler (2016) indicated that the GICC05 counting error underestimates the total uncertainty in some parts of the Holocene

based on the comparison between the radiocarbon dating calibration curve (Reimer et al., 2013, IntCal13) and (Svensson et al., 2008, GICC05), and the work was extended in Adolphi et al. (2018). The objective of this work, however, is to extend GICC05 to the EGRIP core to allow parallel analysis of the records, and we thus refrain from a further discussion of the absolute accuracy of GICC05 here.

## 2.2 Ice-core data sets over the Holocene and last glacial termination

### 2.2.1 EGRIP

Here, we processed and analysed new DEP and ECM records and selected cryptotephra layers in the uppermost 1383.84 m of the EGRIP ice core. At the start of the drilling operation in 2016, the drilling site was located at $75°38'$N and $35°60'$W (see Fig. 1). The average annual accumulation rate is about 100 kg m$^{-2}$ yr$^{-1}$ (equivalent to 0.11 m yr$^{-1}$ of ice) for the period 1607–2011 as determined from a firn core close to the main EGRIP drilling site (Vallelonga et al., 2014). Radar-soundings

suggest the ice thickness to exceed 2550 m and traced radar layers from the NGRIP site suggest that the drill site preserves an undisturbed climatic record of at least 51 kyr (Vallelonga et al., 2014). The camp currently moves about 51 m to the North-Northeast each year (Dahl-Jensen et al., 2019). Fig. 2 presents an overview of the ice-core sections we used in this study. The EGRIP brittle zone is of better quality than the brittle ice from previous Greenland ice core projects such as NEEM and NGRIP. For the EGRIP core, Fig. 3 presents a quality index on the basis of the ratio between validated and total measured DEP and

ECM sample points. This quality index falls below 0.3 between 505 m (4220 a b2k) and 1210 m (11163 a b2k) depth, which is consistent with the brittle zone between 550 m and 1250 m according to the field season reports (Dahl-Jensen et al., 2019). The quality index calculated from the earlier released NGRIP and NEEM DEP data is presented in Appendix A "Quality index for the NGRIP and NEEM ice cores".

### 2.2.2 NGRIP

The GICC05modelext timescale, as discussed in detail in section 2.1 above, is well established for the NGRIP ice core. To fully exploit the potential of DEP records for matching, we processed unpublished DEP data from the NGRIP1 core for the upper part (down to 1298 m), and we used the NGRIP2 (below 1298 m) that was published with the NGRIP ECM data in Rasmussen et al. (2013). The NGRIP1 and NGRIP2 cores have a depth offset of around 0.43 m between corresponding events in the overlapping section (Rasmussen et al., 2013).

 **2.2.3 NEEM**

The firn core NEEM–2008–S1 originates from the NEEM access hole of the main core, drilled during the 2008 field season to a depth of 103 m (Gfeller et al., 2014). We used only ECM data for the matching the upper 100 m, as DEP was not measured on the access-hole core. Below this depth, both DEP and ECM were used to transfer the GICC05 timescale from the NGRIP to the NEEM core (Rasmussen et al., 2013). The shallow and deep cores overlap, forming a continuous record.

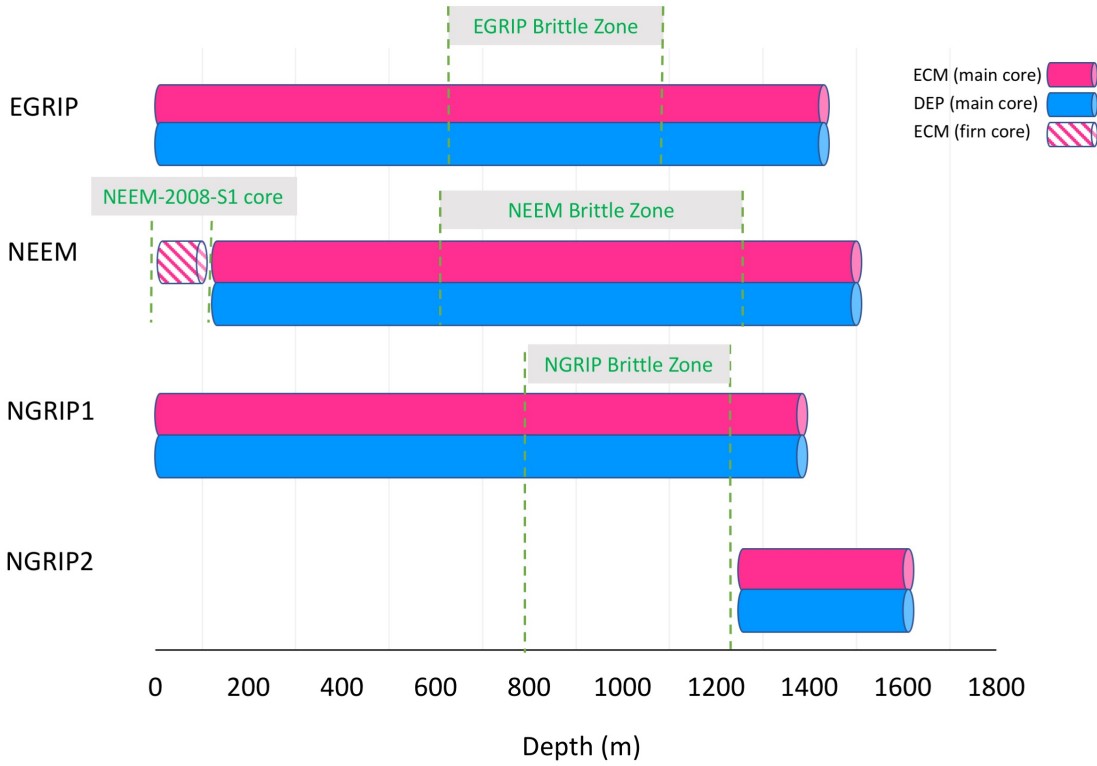

**Figure 2.** Overview of dielectrical profiling (DEP) and electrical conductivity measurements (ECM) that we used for the synchronization between ice cores over the Holocene and late last glacial periods.

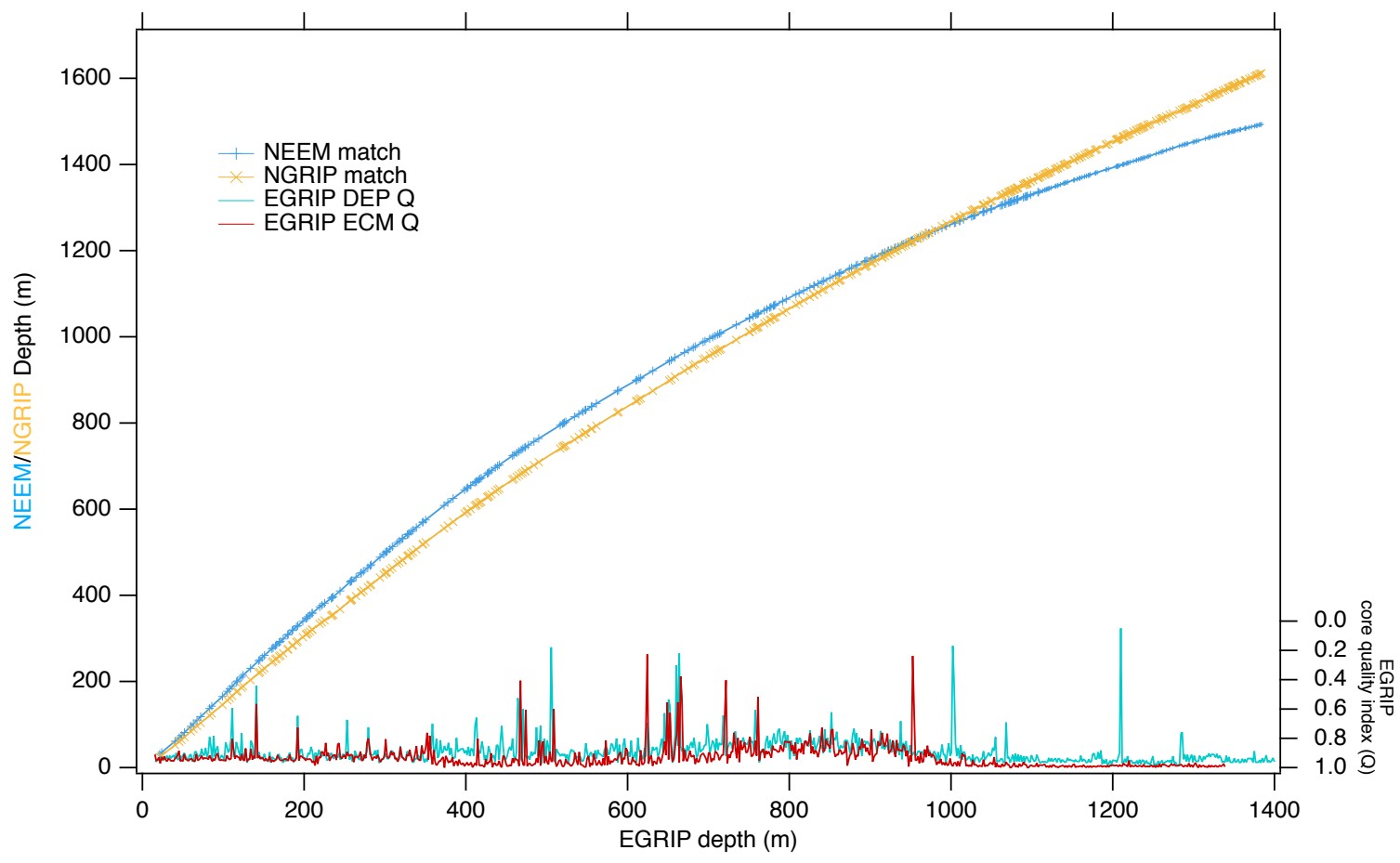

**Figure 3.** Match points between EGRIP, NEEM (blue) and NGRIP (yellow) ice cores based on the DEP and ECM data sets. The core quality index Q as derived from the validated DEP (blue) and ECM (red) data, respectively.

## 2.3 Field measurements and data processing

### 2.3.1 Dielectric profiling (DEP)

Dielectric Profiling (DEP) has been introduced as a system for rapid scanning of ice cores' electric permittivity and conductivity shortly after drilling (Moore and Paren, 1987; Wilhelms et al., 1998). The permittivity and conductivity of ice and firn are determined by their respective densities and conductivities (Wilhelms, 2005). The conductivity is related mainly to acidity, salt and ammonia concentrations of ice cores (Moore et al., 1992, 1994). The dielectric stratigraphy of the EGRIP, NEEM and NGRIP cores were recorded directly during the field seasons with the DEP device described by Wilhelms et al. (1998) (Fig. 4), in the discussion below referred to as "deep-core DEP". DEP is the first measurement within the processing line directly on site. A few minutes before scanning, the core is moved from the core storage to the DEP table. Further along the processing

line, the ice core is split into the different aliquots. For all three ice cores, DEP measurements were carried out on 1.65 m long
sections (Fig. 4d).

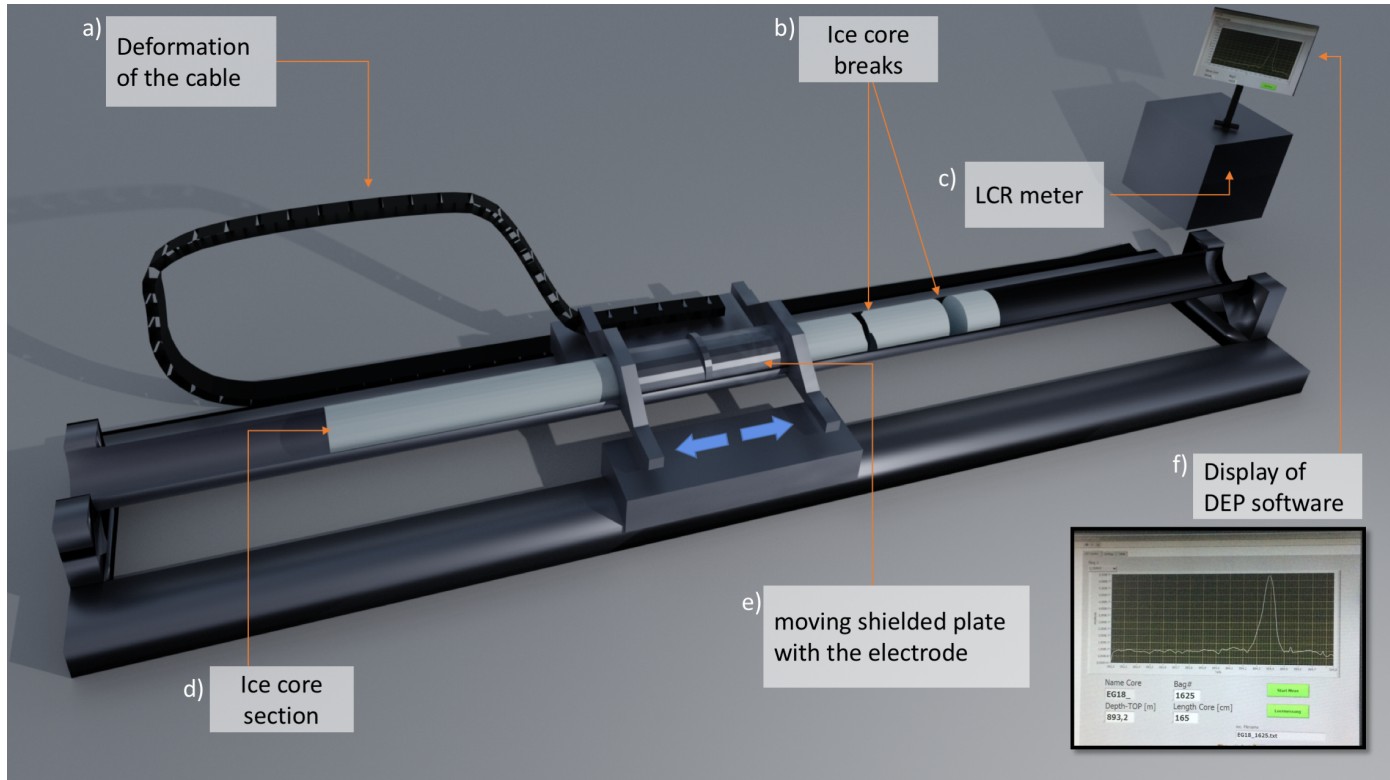

**Figure 4.** Schematic of the DEP instrument.

All cores were scanned with the deep-core DEP device as described in Wilhelms (1996) and Wilhelms et al. (1998) and
the entire scan of a core's section was calibrated with average values of the calibration measurements. For the processing of
the NGRIP and NEEM cores we have improved some features and adapted procedures that were developed for a different
DEP device (Wilhelms, 2000), not yet described elsewhere in the application for the processing of data recorded with the
deep-core DEP device. The identification of peaks is sufficient for the discussion in this paper. However, the data released with
this paper will also be valuable for investigations relying on well-calibrated material properties with absolute calibration, like
e.g. modelling of synthetic radargrams. To provide a comprehensive presentation of the basis of the transfer of the timescale
here, we outline the relevant discussion to operating the DEP system, while the related discussion on precision of the measured
material properties, which are of more relevance for later use of the calibrated data, are presented in Appendix B "Calibration
and corrections to the DEP data".

Due to the drilling procedure and properties of the ice, ice cores can exhibit breaks, broken-off slices or may in some
instances (especially in the brittle zone) be fragmented (Fig. 4b). The missing pieces and free surfaces with possibly high
conductivity have the potential to introduce artefacts into the DEP record. These are clearly identifiable in the permittivity

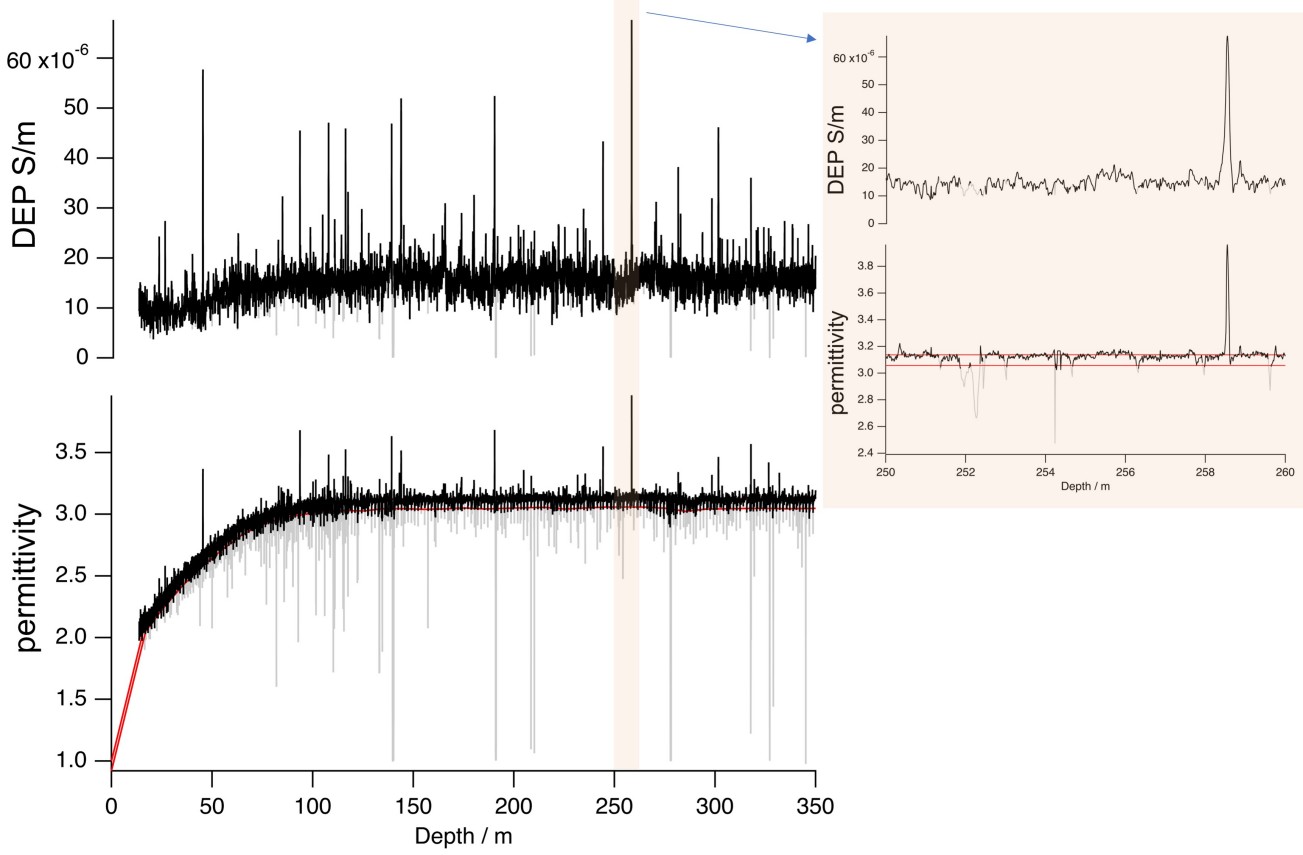

**Figure 5.** Example of DEP data processing (removing core breaks from the raw data). Raw data is plotted in grey and validated data in black. We define insufficient data by permittivity drops below the threshold (indicated by the red line) and reject it during validation. Permittivity is presented in the top and conductivity in the bottom diagram. The insert shows the details for a short section (250–260 m).

record by dropping spikes. For the validation of the data, any drop in permittivity below a certain threshold (cf. the red line
in Fig. 5) identifies a spike to be rejected, where the segment to be rejected is extended from the spike to where the signal approaches the average of the permittivity record. The automated procedure as described in Rasmussen et al. (2013) (section 2.3) is much faster, more consistent in between the three different cores, and has proven to be superior to any approach based on a hand-written protocol, which depends on the judgement of the operator when identifying intervals of bad core. As the permittivity is very sensitive to bad core quality and the conductivity is much less prone to bad core quality, the outlined
validation procedure leads to a robustly validated conductivity record.

The automated validation straight forwardly leads to a definition of a core quality index $Q \in [0, 1]$ by calculating the total length of validated core sections divided by the standard DEP and ECM run lengths of 1.65 m for all three cores. For the EGRIP core, the core quality indices as derived from the DEP and ECM records, respectively, are presented in Fig. 3.

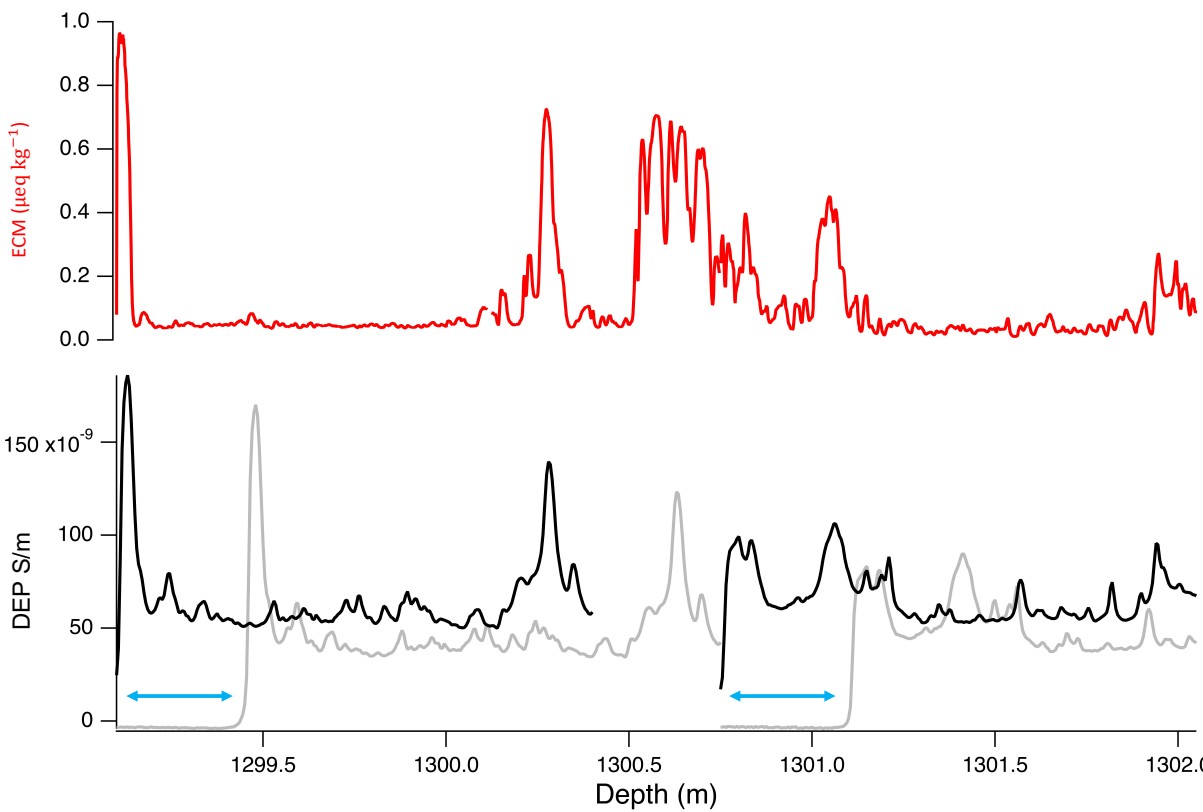

**Figure 6.** Example of the DEP data gap length and relative precision of ECM vs DEP depth assignment. The DEP measurement with a wrong depth in grey and after shifting depth in black. There is a gap of 35 cm (1300.4–1300.75 m) between two DEP measurements. Blue arrows show the early measurements before the starting position of the core. Corresponding ECM data in red.

When processing the EGRIP core, over a certain period the operators erroneously did not reset the starting position of the
scanning electrode of the DEP device. This is clearly identifiable in the records and resulted in recording already ahead of the core's top depth (blue arrows in Fig. 6), then taking measurements over the (correctly set) length of the section, but missing the corresponding length at the bottom end of the core section. The respective core sections have been shifted accordingly, but the missing end sections, which were not recorded, cannot be recovered. The reconstructed DEP record was compared and validated against the ECM record to assign the correct depths. The section about 1285–1385m was corrected in this way,
where in total about 8.5 m of the 100 m were not measured. Furthermore, we relied more heavily on the ECM record than on the DEP record when matching peaks within sections with known problems. Fig. 6 illustrates the corrections in the interval 1299.10–1302.05 m, where a 35 cm data gap between two scanned sections cannot be reconstructed as it was not recorded due to the wrong positioning of the electrode.

### 2.3.2 Electrical conductivity measurements (ECM)

For the EGRIP core, we recorded ECM profiles with the technique described by Hammer (1980) directly in the field. The ECM signal is related to acidity concentrations of ice cores, even with high concentrations of neutral salt (Moore et al., 1992). NGRIP (Dahl-Jensen et al., 2002) and NEEM (Rasmussen et al., 2013) were measured using similar equipment during the respective processing campaigns in the field. For each measurement, the hand-dragged ECM instrument was moved along the depth axis of the ice-core sections' microtome-polished surface (three-bag sections, about 1.65 m long). In order to calibrate

the ECM data, as described in Rasmussen et al. (2013), the ice temperature was measured for each run. In addition, to ensure the ECM quality, we repeated our measurement at least twice for each core section and checked the profiles for the best quality of the measurements. Also, the core-break positions were registered along with measurement by moving the electrodes of the ECM instrument to the respective break position after the core scan, and registering the position in the data file. During the processing, these recorded break marks were used to trim off artefacts and produce the final ECM data set. Data from each day

were calibrated using independent measurements of the physical dimensions of the ECM measurement setup. The first and last few millimetres of recorded data are affected by the proximity to the end of the core and were removed. Areas with dips in the signal around logged core breaks were also muted during processing. Details on the acquisition and processing of the ECM record are laid out in Appendix C "Details on the ECM procedures".

The ECM current, $i$ (in $\mu A$), was converted to ice acidity (in $\mu$equiv. $H^+ kg^{-1}$) by using the relationship $[H^+] = 0.045 \times$
$i^{1.73}$, as suggested by Hammer (1980). Even though conversion from current to acidity and calibration curves have been shown to be ice-core dependent, the matching and synchronization of the ice cores is independent of the absolute values of the calibrated ECM signal as it relies on recognition of similar patterns and peaks in the acidity records (Rasmussen et al., 2013).

The quality of the processed data were checked by comparing independently processed ECM data by three investigators. No major disagreements were found when comparing, and one set of data was agreed on for further use in matching.

### 2.3.3 Tephra horizons

The EGRIP core was continuously sampled for tephra analysis in the field to maximize the probability for detection of volcanic ash deposits, particularly invisible deposits that exhibit low concentrations and/or small grain sizes, known as cryptotephra (Davies, 2015). An aliquot of ice was prepared from the outer curved edge of each 55 cm ice core section and subsampled at 11 cm resolution, providing approximately 30 ml of meltwater per sample. Samples from sections with either significant

peaks (Fig. 4f) in the DEP and ECM or visible layers were separated and prioritized for screening, whereby centrifuged samples were evaporated onto frosted glass microscope slides and covered in epoxy resin to enable scanning by high-magnification light microscopy, as described by Cook et al. (2018). Electron probe micro-analysis (EPMA) by wavelength dispersive spectrometry (WDS) was performed to determine the major element composition of individual grains in each deposit (Hayward, 2012) and EPMA measurements were performed using a Cameca SX100 electron probe microanalyzer at the Tephrochronology

Analytical Unit, University of Edinburgh. This system has five wave dispersive (WD) spectrometers and was calibrated daily using internal calibration standards as described by Hayward (2012). With optimized instrument settings for the analysis of

**Table 1.** Geochemical matches between EGRIP and NGRIP were supported by the similarity coefficient test (SC) of Borchardt et al. (1972) and statistical distance (D2) test of Perkins et al. (1995, 1998). Here we provide SC and D2 values for major elements (normalized to 100%) where 5 major elements (with >%1wt) were used for SC calculations for sample pairs with rhyolitic composition and 7 elements were used for sample pairs with basaltic composition. Values >0.95 suggest products are from the same volcanic source. For D2, seven major elements were used for the comparisons (with >0.01 %wt). The value for testing the statistical distance values at the 99% confidence interval is 18.48 (seven degrees of freedom).

| EGRIP Bag | Depth range | NGRIP /GRIP match | SC | $D^2$ |
|---|---|---|---|---|
| 177 | 96.91–97.02 m | NGRIP 142.61–142.71 m | 0.985 | 1.65 |
| 1627 | 894.41–894.52 m | NGRIP 1163.65–1163.80 m | 0.977 | 4.88 |
| 2094 | 1151.59–1151.70 m | NGRIP 1408.88–1408.89 m (Mortensen et al., 2005) | 0.965 | 5.945 |

small cryptotephra grains (<20 $\mu$m diameter), EGRIP samples were analysed with either a 5 or 3 $\mu$m beam diameter. Secondary standards were analysed to capture instrument drift. The geochemical composition of each layer was compared to deposits in NGRIP and positive matches were used to establish the independent tie-points in between cores listed in Table 1. Major element
biplots (Fig. 7) show graphical correlations for each NGRIP–EGRIP tephra match point, and these are supported by similarity coefficient (Borchardt et al., 1972) and statistical distance ($D^2$) (Perkins et al., 1995, 1998) tests.

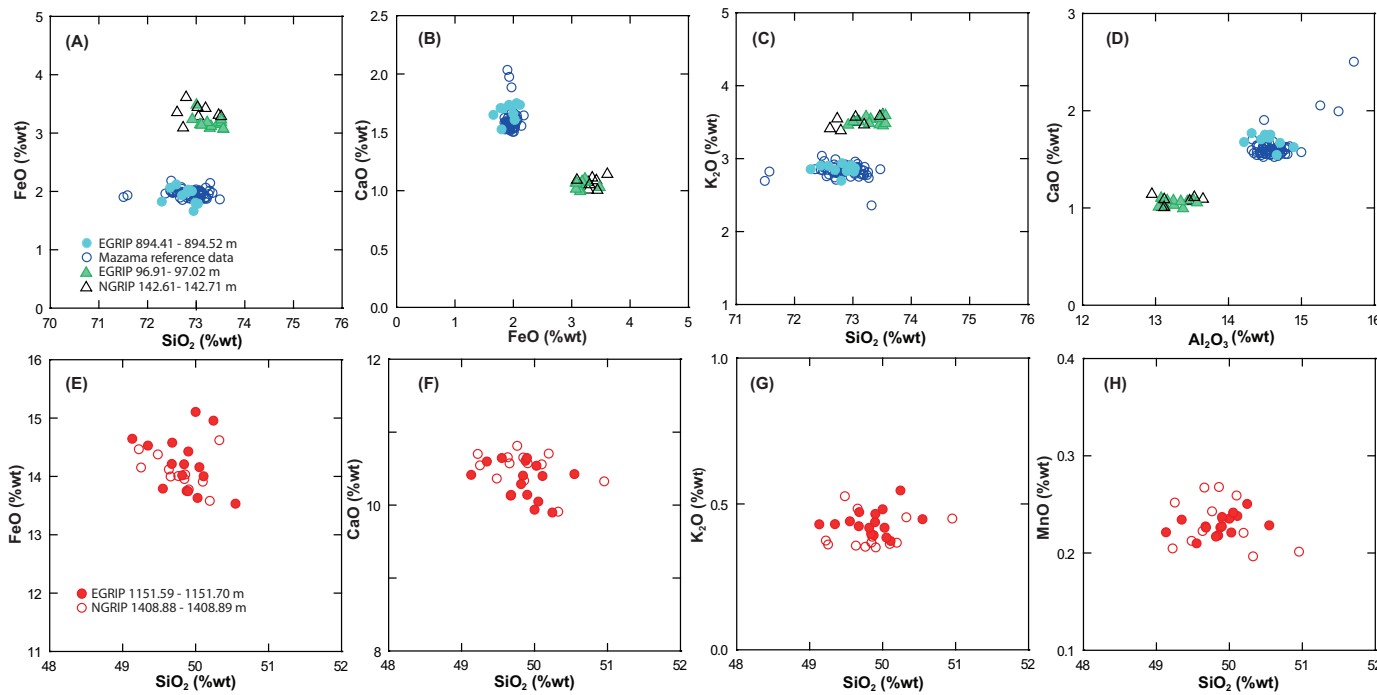

**Figure 7.** Element–element biplots showing geochemical matches between EGRIP and NGRIP samples with the exception of EGRIP 894.41–894.52 m (EGRIP bag 1627), which is shown here with Mazama data from Jensen et al. (2019). Geochemical data are normalised to 100% (anhydrous basis) and analyses with totals below 94 %wt were excluded. Plates (A–D) for EGRIP bags 177 and 1627, Plates (E–H) for EGRIP bag 2094.

## 2.4 Synchronization of dielectric profiling and electrical conductivity measurement records of EGRIP, NGRIP NEEM

Patterns in the DEP records of NGRIP, NEEM and EGRIP were initially matched by one investigator. The same cores' ECM data were matched separately and independently by three different investigators. Both matches are mainly based on clearly identifiable volcanic peaks and also synchronous patterns of other events (see Fig. 8), which not necessarily need to be of volcanic origin, but are assumed to reflect synchronous events. Based on these independent matches, the four investigators identified consistent and reliable common patterns, that are represented in the ECM and/or the DEP records from NGRIP and at least one of the other ice cores. For the confirmation of match points, all records of all three cores were loaded into the Matchmaker tool (Rasmussen et al., 2013) and assessed jointly by all four investigators in the different display options featured by the software. The Matchmaker tool allows easy identification of wrong match points via interactive plots and

on-line evaluation of the match. To validate match points, we plot the depths of the common match points $D_i$ (in EGRIP or NEEM) against $d_i$ (NGRIP). The slope of each of these (depth, depth)-curves is the annual layer thickness ratio of the two cores, $r_i = \frac{D_{i+1} - D_i}{d_{i+1} - d_i}$. Points which deviate from the (depth, depth) curve or create jumps in $r$, are easily recognized and checked again. We only expect significant abrupt changes in $r$ at times where the climate (and thus the relative accumulation rates) shifts due to changes in climate conditions (Rasmussen et al., 2006, 2013; Seierstad et al., 2014; Winski et al., 2019), while the different ice-flow patterns at the cores' sites only lead to slow changes in $r$.

Short-term accumulation variability due to both climatic factors and wind-driven redistribution of snow on the surface can lead to relatively large variations in the ratio of layer thicknesses between different cores, especially when match points are only a few years apart. To reduce short-term accumulation-rate variability in the final timescale, we re-evaluated intervals with large variability in annual-layer-thickness ratios, and removed too closely spaced match points. The final minimum distance between match points is 0.22 m (1206.45 m – 1206.67 m), corresponding to around 3 years. Overall, the match points are reasonably evenly distributed throughout the entire ice core, and the maximum distance between neighbouring match points is 26.6 m (490.06 m – 516.67 m), corresponding to a time interval of 224 years.

## 2.5 Transfer of the GICC05 timescale to the EGRIP ice core

The procedure of transferring the timescale is similar to the approach described in (Rasmussen et al., 2013). Note that we hereby assume that the ratio of annual layer thicknesses is constant between the match points of EGRIP and NGRIP. For each 0.55 m EGRIP depth segment (the so-called *bag*), we obtain the equivalent NGRIP depth by linear interpolation between the depths of the match points $D_i$ in EGRIP and $d_i$ in NGRIP. We then assign a GICC05 age from the annually resolved GICC05 time scale for NGRIP (Vinther et al., 2006; Rasmussen et al., 2006).

The EGRIP timescale inherits the maximum counting error (MCE) from the GICC05 timescale. Our match covers the time period back to 14,967 a b2k where the associated MCE is 196 a. The inaccuracies in the depth registration were estimated by (Rasmussen et al., 2013) to 10 cm ($1\sigma$). For the joint assignment of DEP and ECM patterns, we repeat the assessment by (Rasmussen et al., 2013) for the (EGRIP depth, NGRIP depth) relation by computing the difference $\delta_i$ between each EGRIP match point and the linear interpolated depth derived from the neighbouring points $\delta_i = (D_{i+1} - D_{i-1})/(d_{i+1} - d_{i-1}) * (d_i - d_{i-1}) + D_{i-1} - D_i$, thus merging all match points between EGRIP and NGRIP1 and EGRIP and NGRIP2 into one data set of 377 of the originally 381 match points for further statistical analysis. The difference in number occurs as $\delta$ is not defined for respective start and end points of both respective sequences. The statistical analysis of $\delta$ in Appendix D yields a standard deviation of 0.043 m for the depth assignment of a match point. As the annual layer thickness typically exceeds 0.04 m in the time period considered here, one expects an additional uncertainty for the peak assignment in the order of 1 yr.

Larger errors would occur in the case of erroneously matched sections as discussed in (Rasmussen et al., 2013), but wrong matches are even more unlikely here than in previous work, as three instead of two cores were matched and *ibid.* the authors also point out that erroneously matched sections are particularly relevant for the older part of the core, where the discussion here covers a section with comparably plentiful match points.

### 2.5.1 Precision and accuracy of the time-scale transfer

The central mode of the (depth, depth)-differences $\delta$ as defined above follows a Gaussian normal distribution with a standard deviation of 0.043 m (see Appendix D). This demonstrates that the synchronization ties two cores together at the match points with high precision. Besides the central Gaussian normal distribution, the statistical analysis of $\delta$ identifies an overlaid second Gaussian normal distribution with a standard deviation of 0.19 m. We interpret this distribution as stemming from curvature of the (depth, depth)-curve and as indicative of the average amount of detail, that each point contributes to the description of this curvature. As this difference between the actual match point and the linear interpolation between the neighbouring points depends on the variable curvature of the (depth, depth)-curve, which reflects both accumulation conditions and the evolution of glaciological conditions at both coring sites, the values of $\delta$ will generally be time correlated and cannot be expected to be randomly distributed with depth. We thus estimate that the EGRIP time scale may have time-correlated uncertainties relative to NGRIP of up to a handful of years related to changes in relative accumulation variability and ice-flow conditions not captured by the match points. This uncertainty will be largest in time periods far from the match points and near climatic shifts where the accumulation changed abruptly and not necessarily by the same ratio at different ice coring sites.

Now we assess the combined uncertainties. The GICC05 timescale $t(D)$ inherits the associated maximum counting error from GICC05, and given the analysis of $\delta$ above, we conclude that, at the match points, the time scale is precise relative to NGRIP within about one year ($1\sigma$). However, when we want to know the age at an arbitrary EGRIP depth, additional uncertainties apply due to the interpolation between the match points. There are two dominant sources: As discussed above, variations in relative accumulation rates and ice flow may add up to a handful of years of additional uncertainty relative to GICC05, but there is also a contribution from the choice of interpolation scheme in between the match points. The difference introduced by the choice between the most widely used linear and cubic spline interpolation schemes (Press et al., 1992) is about an order of magnitude larger than the above-mentioned random uncertainty associated with the identification of the match points (see Appendix E).

We maintain linear interpolation for the time scale transfer despite the fact that the slope of the (depth, depth) curve changes instantaneously at the match points. While changes in this slope may in reality occur on many scales due to the intermittency of precipitation, wind-driven redistribution of snow, and relative changes in accumulation rates at the sites, the most significant of these changes are likely to happen at times of climate change rather than at the arbitrary depths of the match points. Considering this, one could consider an interpolation scheme where the change of curvature is distributed over the entire curve and/or concentrated at times of climate changes as derived from the proxies of the ice core. However, we believe that the advantage of obtaining a smoother (depth, depth)-curve does not compare favourably to the additional assumptions needed and the added complexity of the time-scale transfer.

## 3 Results and discussion

### 3.1 Synchronization of the EGRIP, NEEM and NGRIP cores

A total of 257 match points between the EGRIP, NEEM and NGRIP1 ice cores and 124 match points between EGRIP, NEEM and NGRIP2 (total of 381 match points) with an additional three tephra horizons were identified. Fig. 8 shows an example section of ECM/DEP records matched between ice cores. The match points between the ice cores are shown on Fig. 3. In the
245 process of combining match points from all investigators, some match points were removed due to differences in the peak shapes between DEP and ECM data or when there were too many match points very close to each other. There are fewer match points in the interval 600–1100 m due to the brittle zone, which in particular influences the NEEM and NGRIP1 cores. The ECM and DEP do not follow each other closely in the 1245–1283 m interval because of the alkaline nature of the ice associated with stadial conditions in EGRIP. This is due to high dust levels neutralizing the acidity of the ice (Ruth et al., 2003; Rasmussen
et al., 2013).

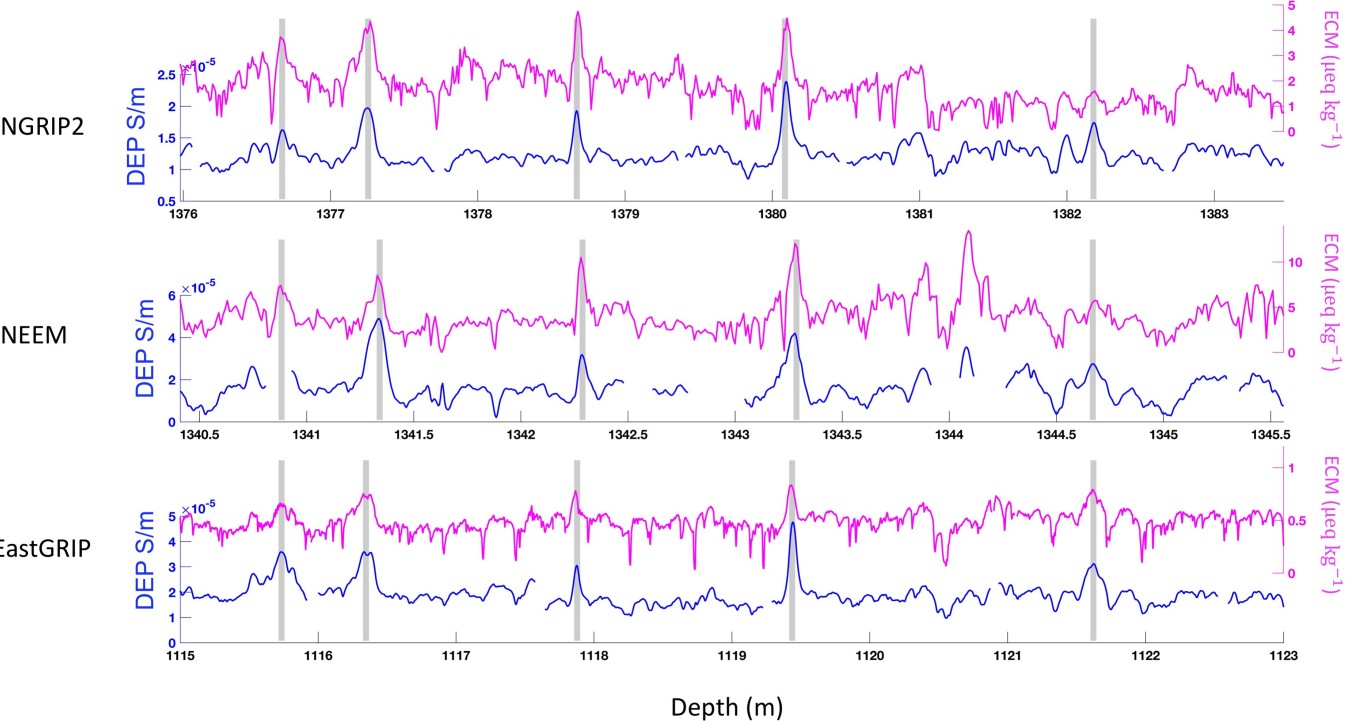

**Figure 8.** Example of the ECM and DEP data match between the NGRIP2 (top), NEEM (middle) and EGRIP (bottom) cores. The match points are marked by grey bands.

## 3.2 Tephra horizons identified for the chronology

Three tephra horizons have been located in EGRIP (Table 1). The locations of these horizons were consistent with the DEP and ECM based synchronization. The tephra horizons thus provide an independent validation for our match points. In addition, ongoing tephra investigations will likely provide additional points for synchronization between ice cores in intervals without DEP and ECM match points.

## 3.3 GICC05-EGRIP-1

As described in section 2.5, the GICC05 (depth, age) timescale was transferred from the NGRIP to the EGRIP ice core based on 381 match points. We name the relationship between depth and age for EGRIP over the Holocene and early last glacial periods GICC05-EGRIP-1 and present the average annual layer thickness between the match points in Fig. 9.

We synchronized the records of the ice cores back to 14.96 ka b2k which corresponds to EGRIP depth 1383.84 m, NEEM depth 1493.29 m, and NGRIP2 depth 1611.98 m. Along with this publication we release a time scale for each 0.55 m section ("bag"). For each EGRIP depth, the corresponding NGRIP depth was found by linear interpolation between the match points, and the GICC05 age was then determined from the published GICC05 time scale for NGRIP. The maximal uncertainty resulting from the choice of interpolation scheme is assessed in detail (see Appendix E1) and is about four years. The relatively smooth (depth, depth) relation of EGRIP–NGRIP and EGRIP–NEEM (see Fig. 3) shows that the ratios of annual layer thicknesses between cores do not vary noticeably between match points. Fig. 10 shows that EGRIP has thinner annual layers than both NEEM and NGRIP ice cores in the upper parts of the cores as also expected from the lower surface accumulation. Ice found in the EGRIP core originates from snow that was accumulating upstream, and accumulation rates increase upstream as the flow line approaches GRIP and NGRIP, where present-day accumulation is about twice of that at EGRIP (Vallelonga et al., 2014; Riverman et al., 2019; Karlsson et al., 2020). Surprisingly, annual layers in EGRIP remain almost constant back to 8 ka b2k (Fig. 9), while the layer thicknesses in large parts of the Holocene part of the NGRIP and NEEM cores thin linearly due to ice flow. We believe that it is a coincidence that the combined effects of the increasing upstream accumulation and flow-induced thinning at EGRIP balance out for the last 8 ka. Despite the lower accumulation at EGRIP, annual layers in EGRIP eventually get thicker than the annual layers in the NEEM and NGRIP ice cores. Below an EGRIP depth of around 700 m, annual layers in EGRIP are thicker than the layers from the same period in the NEEM core, and similarly below 1000 m, EGRIP annual layers are thicker than those in NGRIP (Fig. 10).

There are some gaps in the EGRIP ice-core record due to the brittle zone. However, the smoothness of the depth vs. depth plot in Fig. 3 and the annual layer thickness ratio in Fig. 10 robustly support our time scale based on the match points.

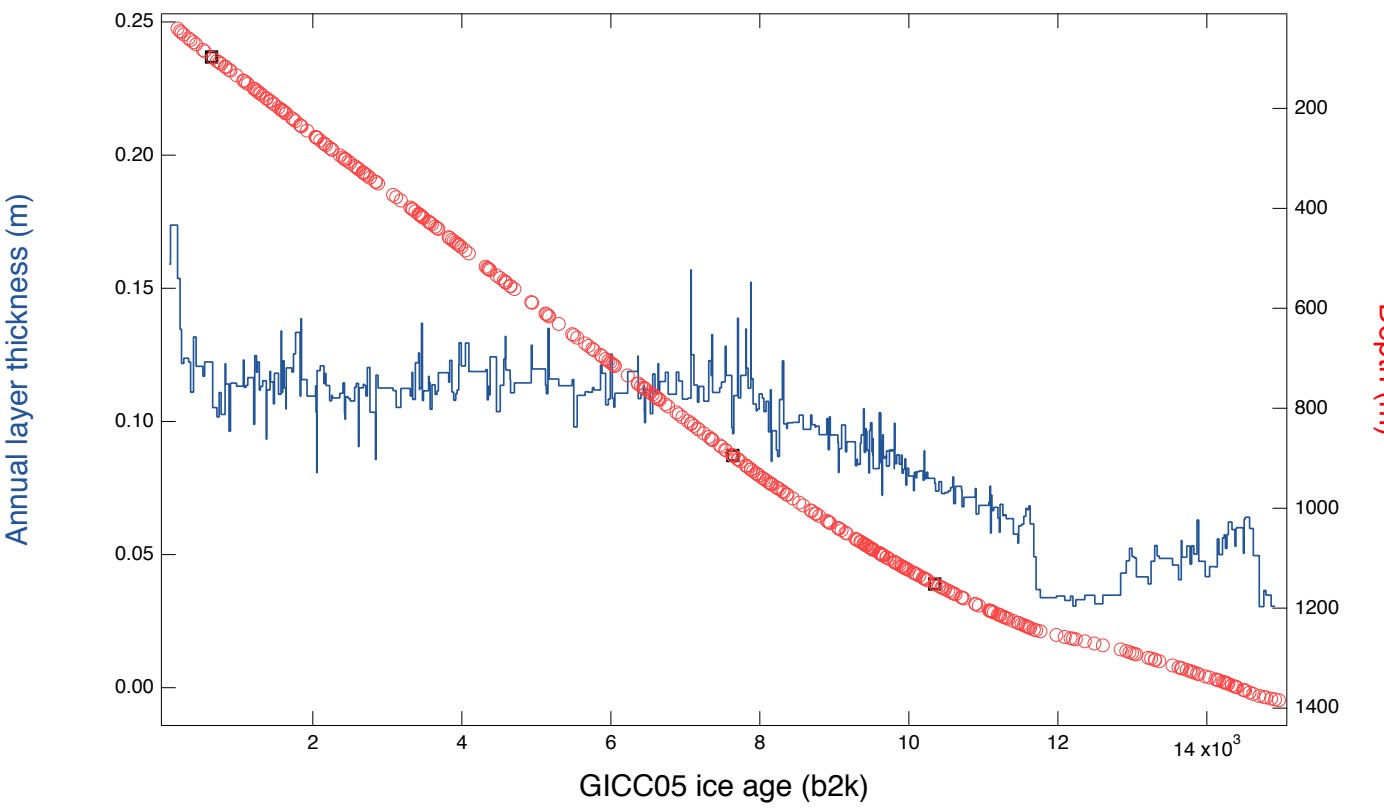

**Figure 9.** Average EGRIP annual layer thicknesses (dark blue line, left y-axis) between the match points. The EGRIP depth–age relationship (right y-axis) with match points (red dots) and the tephra horizons (black squares).

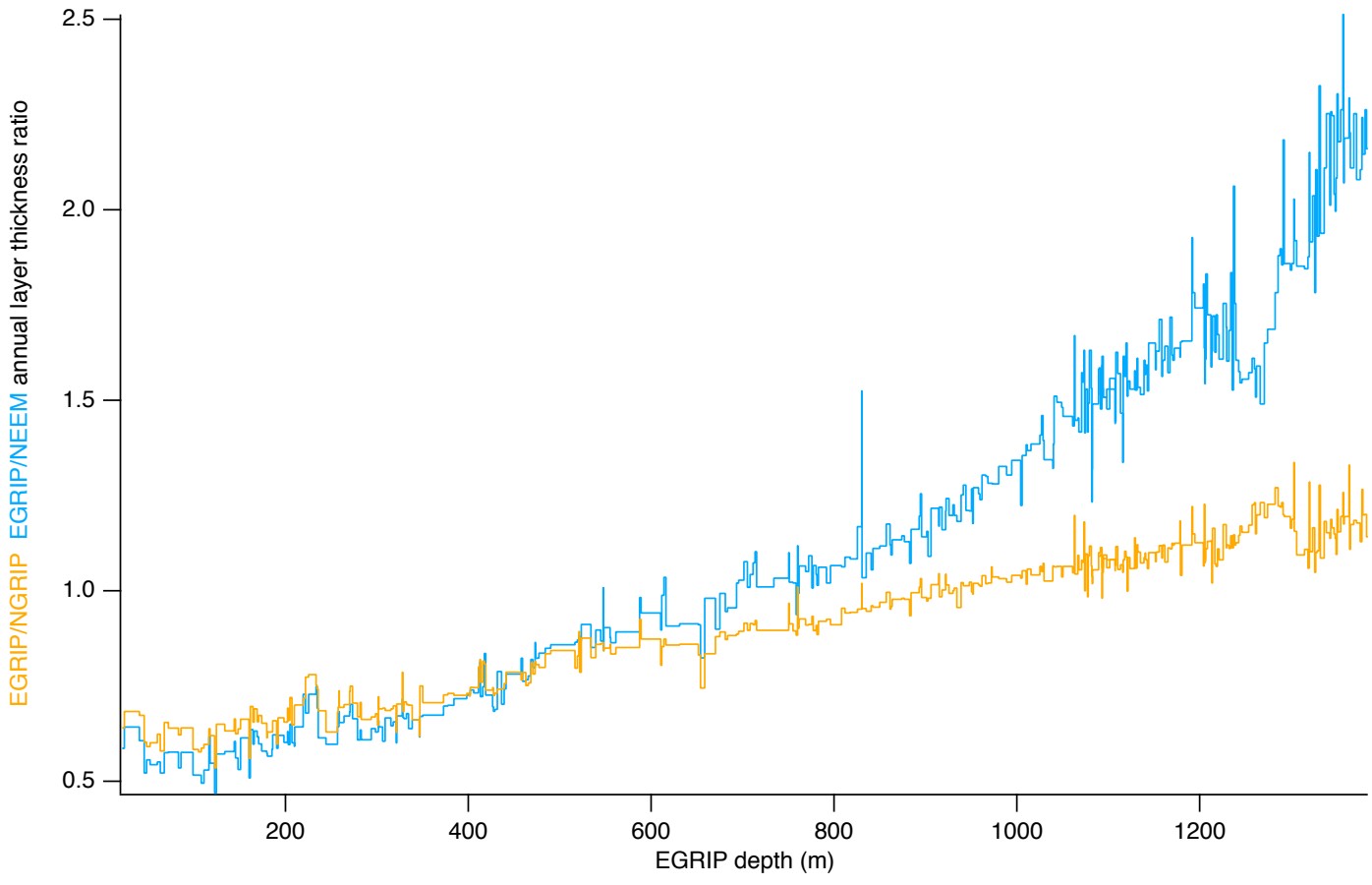

**Figure 10.** The EGRIP/NGRIP (orange) and EGRIP/NEEM (blue) annual-layer thickness ratio (left axis) calculated between neighbouring match points.

### 3.4 Error analysis for the timescale transfer

We statistically treated the "leave-out analysis" ($\delta$), which was already used in previous papers (Rasmussen et al., 2013) to estimate the uncertainty in match points. Binning the values revealed a robust mode that refers the statistical error of the match point assignment. The tails of the distribution relate to more curved intervals of the core. Cubic spline interpolation provides a similar estimate. However, when matching depths, we neither know the true curve nor how big the influence from the interpolation scheme is. We demonstrated that the two fundamentally different interpolation schemes of linear and cubic

spline interpolation give a consistent result. The $\delta$ give qualitatively a similar pattern as the high-resolution difference of the interpolation schemes and has demonstrated its suitability to estimate the uncertainty of the interpolation scheme. They deviate more than the statistical uncertainty of the match-point assignment, but this is less than the tenfold of the statistical error for peak identification and for the timescale here it introduces an error of up to 4 years, while the inherited maximum counting error of the timescale increases from about 1 year to 2 centuries over the matched record.

## 4 Conclusions

We have established the initial chronology for the EGRIP deep ice core in Greenland which encompasses the Holocene and late glacial periods. We have established the depth–age relation for the upper $\sim$ 1383.84 m of the core back to approximately 14.96 ka b2k based on the GICC05 time scale and labelled it GICC05-EGRIP-1. After field measurements and processing of the ice-core data, we relied on the DEP and ECM records for the synchronization, using 381 match points between EGRIP, NEEM and NGRIP ice cores. The identification of tephra match points between the EGRIP and NGRIP core provide an independent tool for validating this synchronization. We used the ratio of annual layer thickness between ice cores as a tool to evaluate our match points. This first timescale can help to interpret, design sampling strategies and improve the understanding of the forthcoming EGRIP data sets.

## 5 Supplementary data

With the final version of this paper we will publish the following data sets at www.pangaea.de and www.iceandclimate.dk/data:

– GICC05-EGRIP-1 time scale for the EGRIP ice core (Mojtabavi et al., 2020a, https://doi.pangaea.de/10.1594/PANGAEA. 922139).

– Specific conductivity measured with the dielectric profiling (DEP) technique on the EGRIP ice core, 13.77-1383.84 m depth (Mojtabavi et al., 2020b, https://doi.pangaea.de/10.1594/PANGAEA.919313).

– Permittivity measured with the dielectric profiling (DEP) technique on the EGRIP ice core, 13.77-1383.84 m depth (Mojtabavi et al., 2020c, https://doi.pangaea.de/10.1594/PANGAEA.922138).

– Acidity measured with the Electrical Conductivity Method (ECM) on the EGRIP ice core (down to 1383.84 m depth), converted to hydrogen ion concentration (Mojtabavi et al., 2020d, https://doi.pangaea.de/10.1594/PANGAEA.922199).

– Specific conductivity measured with the dielectric profiling (DEP) technique on the NEEM ice core (down to 1493.297 m depth) (Mojtabavi et al., 2020e, https://doi.pangaea.de/10.1594/PANGAEA.922193).

– Permittivity measured with the dielectric profiling (DEP) technique on the NEEM ice core (down to 1493.297 m depth) (Mojtabavi et al., 2020f, https://doi.pangaea.de/10.1594/PANGAEA.922195).

– Specific conductivity measured with the dielectric profiling (DEP) technique on the NGRIP1 ice core (down to 1372 m depth) (Mojtabavi et al., 2020g, https://doi.pangaea.de/10.1594/PANGAEA.922191).

– Permittivity measured with the dielectric profiling (DEP) technique on the NGRIP1 ice core (down to 1372 m depth) (Mojtabavi et al., 2020h, https://doi.pangaea.de/10.1594/PANGAEA.922192).

## Appendix A:  Quality index for the NGRIP and NEEM ice cores

For the NEEM and NGRIP ice cores we calculated similar quality indices as provided for EGRIP above. They are presented together in Fig. A1.

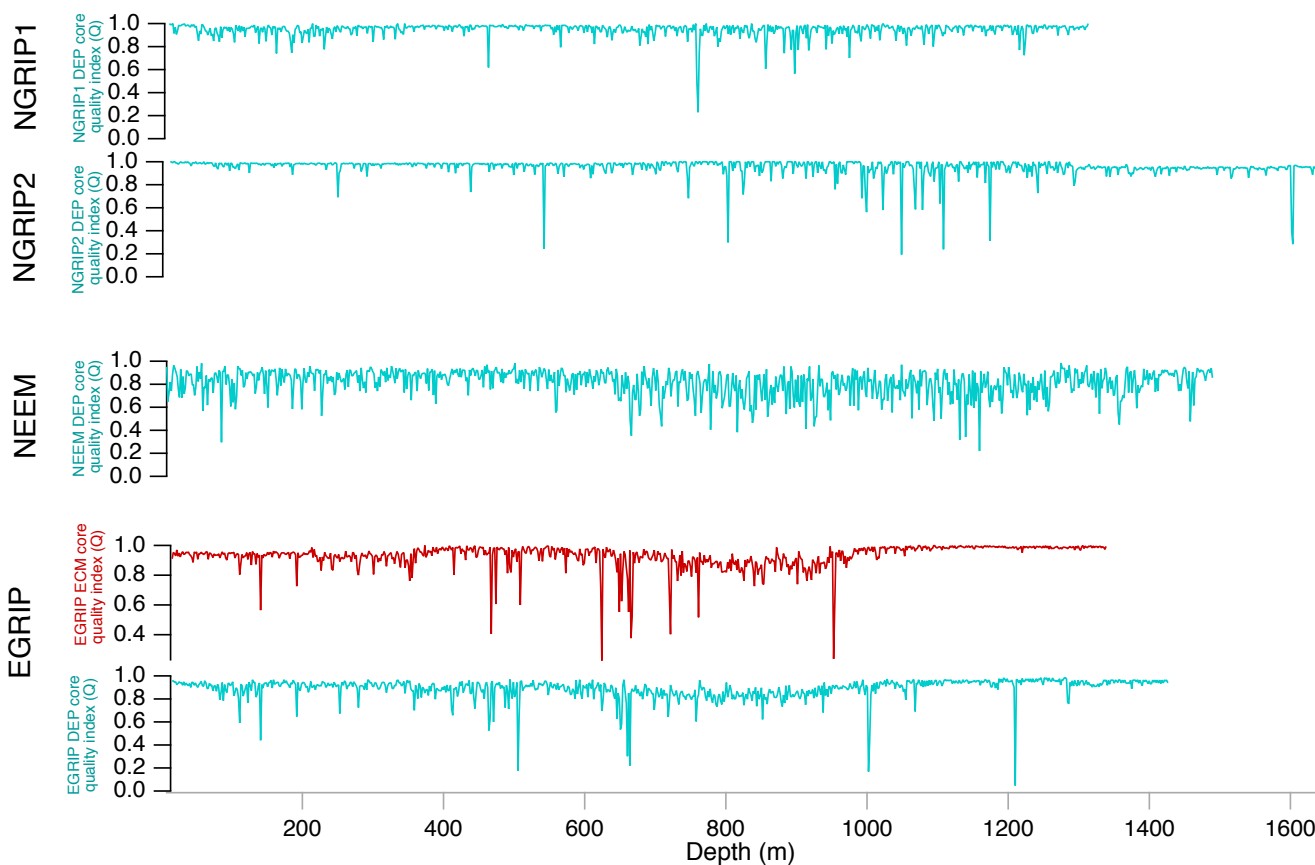

**Figure A1.** Quality indices for the EGRIP, NGRIP and NEEM ice cores.

## Appendix B:  Calibration and corrections to the DEP data

As a correction in the few percent range we correct the offset, introduced by the changing stray admittance due to the varying cable geometry due to their movement during the measurement, by subtracting the course of free-air measurements from the respective measurement of a core section along the DEP device when processing the data. This is possible as the stray admittance is connected in electric series and is thus additive. This measure reduces coherent noise on the record, which is e.g. important when using the records for spectral analysis.

For the processing of the two NGRIP cores, reproducibility was ensured by laying the cables out to move freely in the same way for all measurements in between two recorded free-air measurements. This was improved for the NEEM and EGRIP processing by placing the cables into cable channels that enforce repeatable deformation.

For the calibration of the DEP device, free-air measurements without ice were recorded frequently, usually at least twice daily before processing started and finished. The slight capacitance and conductance variation on the order of less than 4 fF and 500 pS, thus corresponding to relative permittivity changes of 4 fF/63 fF = 0.06 and conductivity changes of (500 pS)/(63 fF)*(8.8542 pF) = 70 nS, along the DEP device is due to the unavoidable deformation of the cables (Fig. 4a) when moving the scanning electrode along the device (Fig. 4e)." Compared to the properties of pure glacier ice (ref. to Fig. 5) these variations are in the order of 2% for the permittivity and 5‰ for the conductivity. Additionally, an offset of few nS residual conductance may remain even after performing the correction routines of the LCR meter (inductance L, capacitance C, resistance R) bridge (Fig. 4c).

As a further correction in the few percent range, we developed procedures to determine the true absolute free air-capacitance of the DEP capacitor. This is relevant when determining calibrated absolute values of the material properties: permittivity and conductivity. In the original publications (Wilhelms, 1996; Wilhelms et al., 1998) the proper calibration of the device is cross checked by comparison with the theoretical capacitance value of 63.4 fF, where for a precisely adjusted DEP bench, the free air capacitance coincides with in less than 2 fF. These small deviations from the theoretical value might well be due to mechanical tolerances like the electrode length in the range of a few tenths of a mm. However, for a slightly differently adjusted device (e.g. one with slightly more clearance to the core), the deviation from the ideal value of the free air capacitance might be a few fF more. Besides the calibration uncertainty of the LCR meter, it might also include a component of cable stray capacitance, which is not identifiable in the LCR meter's automated correction procedure.

To even proceed from the correction of cable stray admittance variation along the course of the device by simple subtraction towards absolute precision one needs to know the free-air-conductance, which is expected to vanish for the empty device, and the true free-air-capacitance of the capacitor, which needs a special approach as it cannot be measured directly and separately from other interfering capacitances.

In parallel to the NGRIP project, Wilhelms (2000) developed a calibration procedure for a custom DEP device with fixed electrodes, which was optimised for firn studies to establish the DECOMP model (Wilhelms, 2005). For the EGRIP core processing, we transferred the principle of introducing concentric metal tubes to the deep-core DEP bench and upgraded it with a rack to mount to move the tubes of different diameters along with DEP electrodes and record the capacitance along the DEP device. The tube in the electric field increases the capacitance of the arrangement and Wilhelms (2000) derives the theory to calculate the effective relative permittivity of the setup. For the calibration, tubes with radii in approximately 10 mm increments between 0 and 70 mm represented effective permittivity standards ($|\varepsilon|$) between 1 to 4 (Fig. B1). The result is a calibration curve which holds for the calculation of a consistent free-air-capacitance for the correction of the DEP measurement of the EGRIP core. The free-air-capacitance is the proportionality factor of the measured capacitance and the effective permittivity, i.e. the slope of the graph in Fig. B1. From this analysis one derives the true free-air-capacitance of the deep-core DEP device as adjusted when it was assembled.

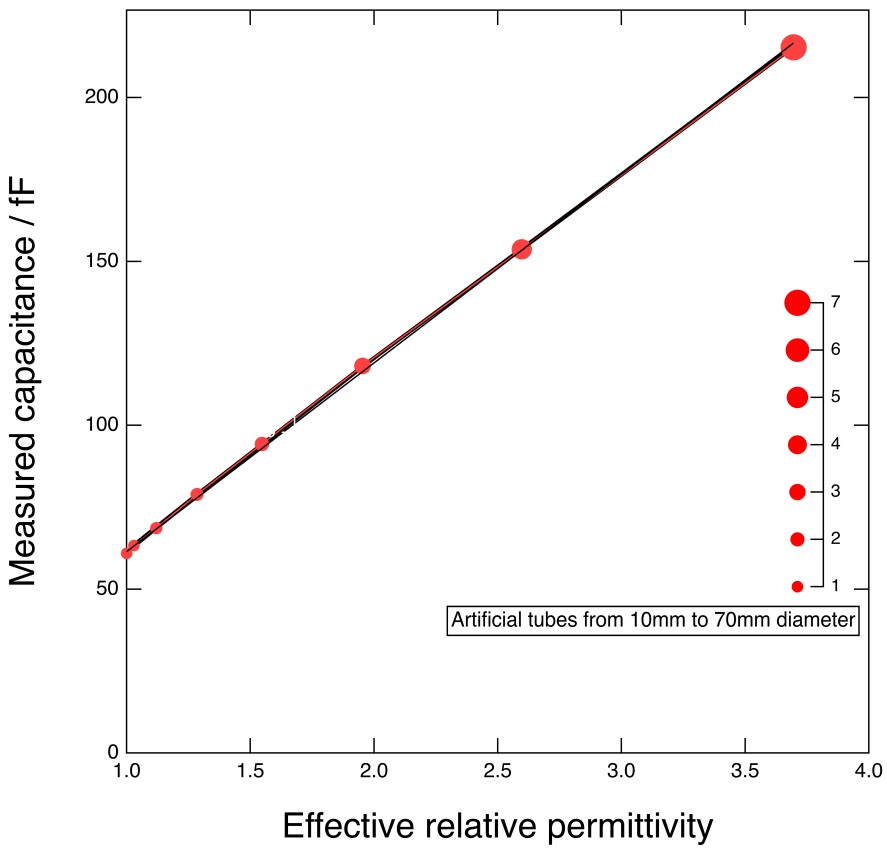

**Figure B1.** Calibration curve of the measurements with artificial tubes and free-air measurement for DEP device.

Now, we know the true free-air-capacitance (as determined from the just outlined calibration) together with the true (vanishing) conductance for an empty device. The free-air empty capacitor measurements thus determine the (additive) stray conductance and capacitance, which is the difference of the measurement and the reference values. The derived stray conductance and capacitance profile along the course of the DEP device are then subtracted from the actual ice-core measurements. The calculation of both material properties involves a division with the free air-capacitance, where introduction of further errors is minimised by using the precise value from the calibration procedure as outlined above.

Precise permittivity and resultingly precise free air capacitance values mainly make a difference when e.g. computing the wave propagation speed of radar waves while modelling synthetic radargrams (Eisen et al., 2006). For NGRIP and NEEM we did not yet perform the calibration procedure. From assigning the two-way travel time of prominent radar reflectors to volcanic spikes in the core, one can determine the radar wave propagation speed and calculate the ice's permittivity. As the NGRIP and NEEM datasets will also be used for comparison with radar surveys later, we determined the free air capacitance by averaging

the measured capacitance over deep core sections and dividing with the expected permittivity of ice of $3.15 \pm 0.1$ which computes the free air capacitance with 3% relative error, which is only about 2 fF absolute error for the free air capacitance.

To sum up, all material properties' datasets we derived here are only subjected to a few percent absolute error. While the EGRIP record is calibrated independently, for the NGRIP and the NEEM cores the permittivity, as determined from radar wave propagation, was used. The latter is a very minor restriction as this cannot be checked independently, but is not of practical relevance.

Due to the varying temperature in processing area throughout the field seasons, the core was not processed at a consistent temperature and we don't have the temperature readings avail to provide consistently harmonized conductivity data. The missing temperature correction does not affect the use of conductivity peaks for synchronization purposes in between ice cores, which is relevant for the discussion here. When e.g. deriving radar wave absorption coefficients from the presented conductivity record, one would have to be very cautious and have this limitation of the data in mind.

## Appendix C: Details on the ECM procedures

The depth scale of the ECM profile was assigned based on the recorded movement of the electrodes interpolated between the logged top and bottom depth of individual ice-core sections. To investigate the quality of the depth assignment, a bag mark position analysis was carried out on the section below the brittle zone in EGRIP, ~1160–1760 m. Only ice core sections with an undamaged core of 1.65 m length were included in the analysis. Each 1.65 m section contains the equivalent of three 0.55 m bags, and the true depth of the bag interfaces ("bag marks") separating the first and second bags and the second and third bags, respectively, are known. During ECM measurements, these bag marks are logged (just as the break marks), but are not used for the processing. After processing of the ECM signal, the position of logged bag marks were interpolated onto the same depth scale as the processed ECM signal, making it possible to compare the true depth of these marks to their depths in the processed data. The distance in depth between logged and expected positions of individual bag marks were calculated for all sections included in the analysis. It was found that the depth assignment of the bag marks were almost always accurate within 20 mm, with mean distance $\mu = 8.3$ mm and standard deviation $\sigma = 7.9$ mm.

## Appendix D: Statistical analysis of the (depth, depth)-match with linear interpolation

Each match point's depth assignment has an uncertainty due to the varying peak form, which is caused by regional deposition differences and short-term accumulation variations. This peak assignment uncertainty dominates the distribution of $\delta$ in sections where the EGRIP–NGRIP1/2 depth curve is straight. However, varying conditions at the time and place of the snow deposition or different ice flow pattern between the ice cores cause differently evolving annual layer thickness ratios $r_i = \frac{D_{i+1} - D_i}{d_{i+1} - d_i}$ and recognizable curvature of the (depth, depth)-curve. For the distribution function of $\delta$, we expect a normal distribution of the peak assignment overlaid by a distribution from the curvature of the (depth, depth) curve, in the following referred to as "the refinement".

For further statistical treatment, we bin the $\delta$ values. As $\delta \in [-0.42929\text{m}, 0.385968\text{m}]$, we counted the occurrence $N(i)$ of values for the $i = 0 \ldots 20$ intervals $[-(0.525 + i * 0.05)\text{m}, (-0.475 + i * 0.05)\text{m})$ in between -0.525 m and 0.525 m and display the data in the following histogram (ref. Fig. D1). The standard deviation of the counts is $\sigma(N) = \sqrt{N}$. $n = 14$ bins are occupied. The distribution of $\delta$ is presented in Fig. D1.

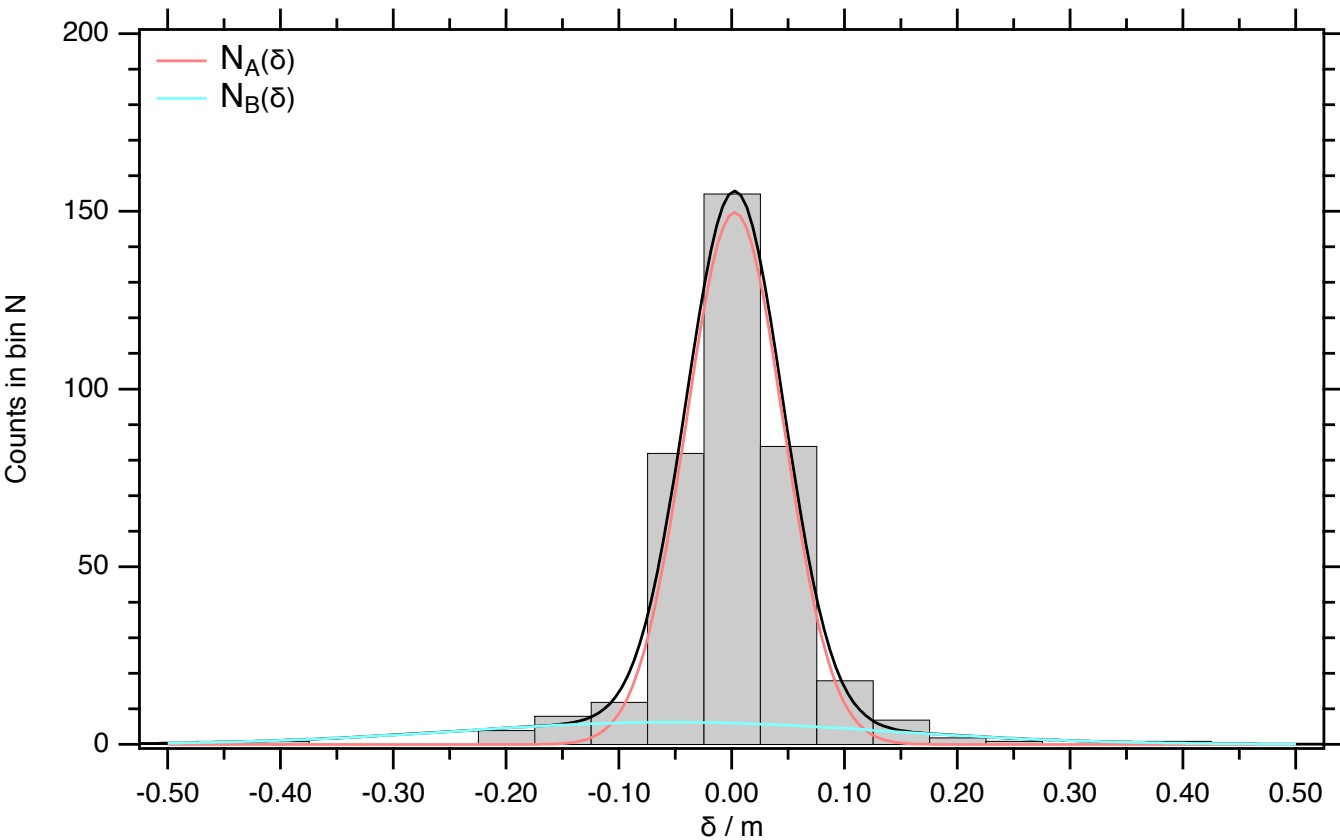

**Figure D1.** Histogram of the difference data with linear interpolation $\delta_i = (D_{i+1} - D_{i-1})/(d_{i+1} - d_{i-1})(d_i - d_{i-1}) + D_{i-1} - D_i$ for the match points between EGRIP–NGRIP1/2.

The Gaussian normal distribution for the peak assignment in the centre seems to be overlaid by a second Gaussian normal distribution representing the refinement of the depth scale. This is indicated by the quite wide tails of the distribution. Following the just posed assumption, that the refinement distance is statistically normal-distributed $N_B(\delta) = B \exp(-(\delta - \mu_B)^2/\sigma_B^2/2)/\sqrt{2\pi}/\sigma_B$ and that the uncertainty of the peak assignment in smooth intervals is normal-distributed $N_A(\delta) = A \exp(-(\delta - \mu_A)^2/\sigma_A^2/2)/\sqrt{2\pi}/\sigma_A$, we $\chi^2$-fitted the sum $N(\delta) = N_A(\delta) + N_B(\delta)$. The weight of the $\chi^2$ is the counting error $\sigma(N) = \sqrt{N}$. The fit of the $c = 6$ independent parameters converged and yielded a $\chi^2 = 4.7$ for the $f = n - c = 14 - 6 = 8$ degrees of freedom, indicating that the fitted distribution is supported by the data ($\int_{4.7}^{\infty} \chi^2(8) = 0.79$).

The $\chi^2$-fit computes for the peak assignment distribution $N_A(\delta)$ a scaling factor $A = (16 \pm 1)$m, the shift from the centre
$\mu_A = (0.003 \pm 0.003)$m and a standard deviation $\sigma_A = (0.043 \pm 0.002)$m. Similarly for the refinement distance $N_B(\delta)$: $B = (3 \pm 0.7)$m, $\mu_B = (-0.05 \pm 0.04)$m and $\sigma_B = (0.19 \pm 0.04)$m.

Solving $N_A(\delta_j) = N_B(\delta_j)$ for the roots $\delta_1 = -0.11m$ and $\delta_2 = 0.12m$ defines the inner interval that is dominated by the peak assignment statistics. 349 points are in the interval $(\delta_1, \delta_2)$ and the direct statistical evaluation confirms $\sigma_A = 0.043m$ (in the main paragraphs of the paper we label this as statistical error for the peak assignment $\Delta D$) and $\mu_A = 0.003$, where the skew (0.06) and the kurtosis (-0.05) are small and support normal distribution $\delta \in (-0.11\text{m}, 0.12\text{m})$. A Shapiro Wilk test confirms normal distribution of the peak assignment errors, as $W = 0.996$ and the corresponding p-Value $p = 0.54$.

As the refinement distance is overlaid by the peak assignment statistics in the centre of the distribution, none of the standard statistics is applicable, but the $\chi^2$-fitted $N_B(\delta_i)$, can be $\chi^2$-tested for the refinement-distance dominated bins. $\chi^2 = 4.0$ over the bins $i = 1, 2, 6, 7, 13, 14, 15, 17, 18$ in the tail that at most are marginally influenced by $N_A(\delta_i)$. The $n = 9$ bins together with initially $c = 3$ fitted parameters computes $f = n - c = 6$ degrees of freedom, which supports the refinement distance being normal distributed ($\int_{4.0}^{\infty} \chi^2(8) = 0.68$).

## Appendix E: Statistical analysis of the (depth, depth)-match with cubic spline interpolation

Here, we quantify the difference between using linear interpolation and interpolation by cubic splines, which is a widely used scheme. The latter has the benefit of using smooth curves, such that the (depth, depth) curve and its derivatives are continuous, but as discussed above, several factors may cause the real (depth, depth) curve to be non-differentiable or even discontinuous, and we therefore maintain our practice of linear interpolation between the depths of the match points.

Analog to the definition of $\delta$ we define $\Sigma_i = S(D_1, \ldots, \cancel{D_i}, \ldots, D_n) - S(D_1, \ldots, D_i, \ldots, D_n)$, where $S(D_1, \ldots, D_i, \ldots, D_n)$ is a cubic spline calculated for all match points and $S(D_1, \ldots, \cancel{D_i}, \ldots, D_n)$ a cubic spline calculated for all but the i-th match point.

Analog to the above analysis with linear interpolation, we expect that each match point's depth assignment has an uncertainty due to the varying peak form, which is caused by regional deposition differences and short-term accumulation variations. This peak assignment uncertainty determines the distribution of $\Sigma$ in sections where the EGRIP–NGRIP1/2 depth curve is straight and no systematic glaciological differences occur. However, varying accumulation conditions at the time and position of snow deposition or different ice flow patterns influencing the ice cores cause differently evolving annual layer thickness ratios $r_i = \frac{D_{i+1} - D_i}{d_{i+1} - d_i}$ which leads to curvature of the (depth, depth) curve. This curvature may not be captured if there are no match points at the relevant depth, and different interpolation schemes will make different predictions across intervals without match points. For the distribution function of $\Sigma$ we thus expect two contributions: One from the uncertainty of peak assignment and the other one from the refinement.

For further statistical treatment, we bin the $\Sigma$ values. As $\Sigma \in [--0.674604\text{m}, 0.364446\text{m}]$, we counted the occurrence $N(i)$ of values for the $i = 1 \ldots 21$ intervals $[-(0.725 + i*0.05)\text{m}, (-0.675 + i*0.05)\text{m})$ in between -0.675 m and 0.375 m and

display the data in the following histogram (ref. Fig. E1). The standard deviation of the counts is $\sigma(N) = \sqrt{N}$. $n = 17$ bins are occupied. The distribution of $\Sigma$ is presented in Fig. E1. The weight of the $\chi^2$ is the counting error $\sigma(N) = \sqrt{N}$.

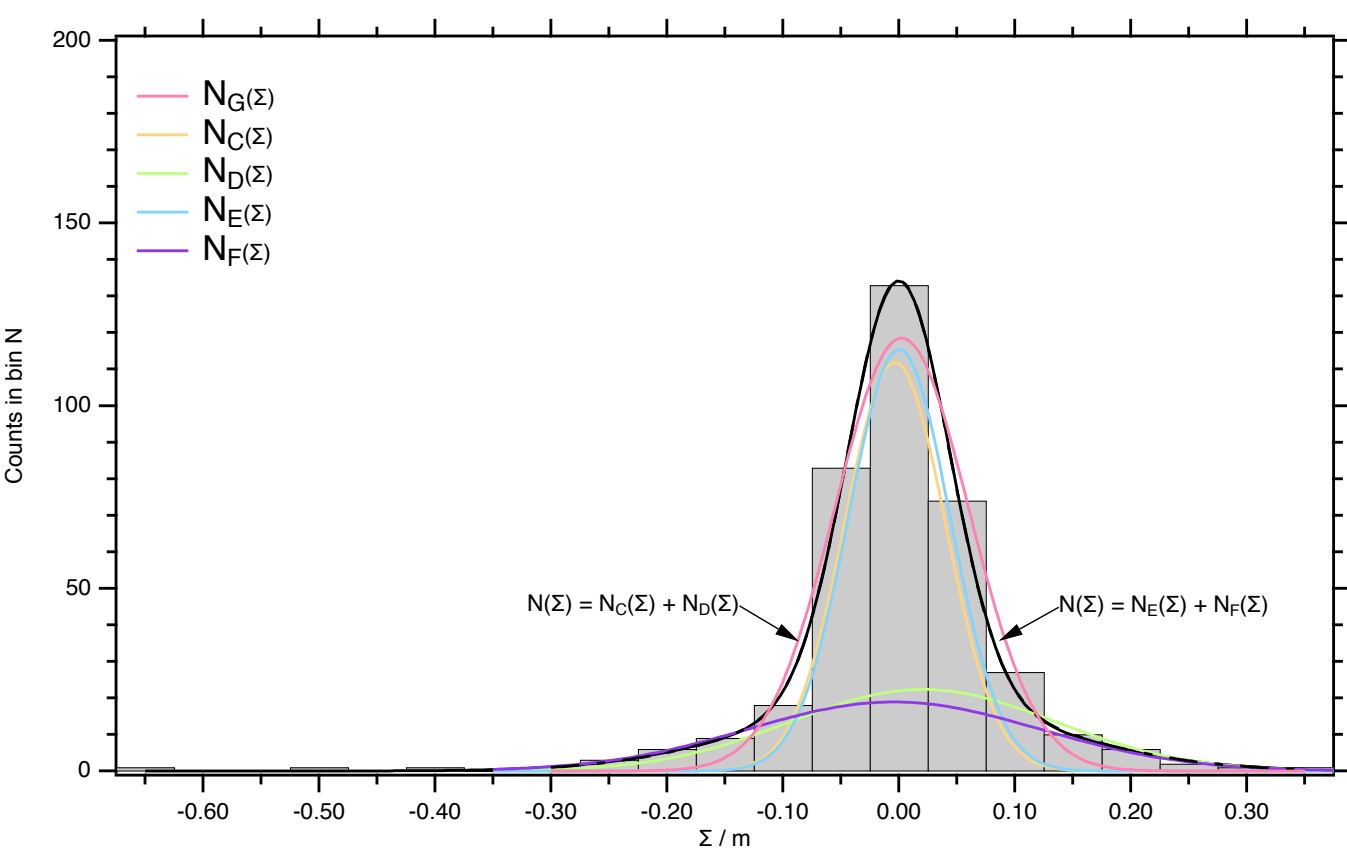

**Figure E1.** Histogram of the difference data with cubic spline interpolation $\Sigma_i = S(D_1, \ldots, \cancel{D_i}, \ldots, D_n) - S(D_1, \ldots, D_i, \ldots, D_n)$ for the match points between EGRIP—NGRIP1/2.

Like for the (depth, depth) differences for the linear interpolation $\delta$, the distribution of $\Sigma$ appears having too wide tails to match a Gaussian distributions. We support this by trying to fit a single Gaussian normal distribution $N_G(\Sigma) = G \exp(-(\Sigma - \mu_G)^2/\sigma_G^2/2)/\sqrt{2\pi}/\sigma_G$. to the data. Even when treating the 3 values below $\Sigma < 0.325$ as outliers and restricting the fit to the $n = 14$ bins around 0, we minimize $\chi^2 = 30.8$. For the $c = 3$ fitted constants, the degree of freedom is $f = 11$. This indicates that the fitted distribution is not supported by the data ($\int_0^{30.8} \chi^2(11) = 0.99$).

The Gaussian normal distribution for the peak assignment in the centre seems – as for the $\delta$ – to be overlaid by a second Gaussian normal distribution representing the refinement of the depth scale. Following the just posed assumption that the refinement distance is statistically normal-distributed $N_D(\Sigma)$ and that the uncertainty of the peak assignment in smooth intervals is normal-distributed $N_C(\Sigma)$, we tried to $\chi^2$-fit $N(\Sigma) = N_C(\Sigma) + N_D(\Sigma)$ to the entire dataset with $c = 6$ fitting parameters.

For the $n = 17$ bins of the entire dataset $\Sigma$ we minimised $\chi^2 = 48.7$, which suggests the model does not describe the data (for details refer to Fig. E1, where the figures for a similar treatment as for rejecting $N_G(\Sigma)$ are provided).

By just treating the 3 values (out of 377 in total) below $\Sigma < 0.325$ as outliers, which is justified, when we are mainly interested in assessing the central part of the distribution, which refers to the match-point assignment. Thus, repeating the fit for the bins $i = 8, /ldots, 21$ ($n = 14$) an labelling the fitted function $N(\Sigma) = N_E(\Sigma) + N_F(\Sigma)$ to clearly distinguish the result from the fit to the entire dataset before, we are able to minimize $\chi^2 = 3.09$ for the $f = n - c = 11$ degrees of freedom, indicating that the fitted distribution is supported by the data ($\int_{3.1}^{\infty} \chi^2(11) = 0.93$).

The $\chi^2$-fit computes for the peak assignment distribution $N_E(\Sigma)$ a scaling factor $E = (12.6 \pm 1.6)$m, the shift from the centre $\mu_E = (-0.0003 \pm 0.004)$m and a standard deviation $\sigma_E = (0.044 \pm 0.004)$m. Similarly for the refinement distance $N_F(\Sigma)$: $F = (6.0 \pm 1.5)$m, $\mu_F = (0.004 \pm 0.013)$m and $\sigma_F = (0.13 \pm 0.02)$m.

For the linear interpolation scheme (ref. to Appendix D) the peak assignment contribution ($N_A(\delta)$ dominates the central bins. Only 4% of the counts are attributed to the refinement ($N_B(\delta)$) and we could define the interval where to perform a direct statistical analysis of the $\delta$ values for the peak assignment mode from the intersections of $N_A(\delta)$ and $N_B(\delta)$. For the cubic spline interpolation scheme 16% of the counts are contributing to the refinement $N_F(\Sigma)$ in the central bins. Thus the roots $\Sigma_1 = -0.089m$ and $\Sigma_2 = 0.087m$ of $N_E(\Sigma_k) = N_F(\Sigma_k)$ do not define the complete inner interval, where the peak assignment contributes and it does not as clearly dominate the distribution of $\Sigma$. The peak assignment and the refinement modes do not separate as clearly as for the linear interpolation, and a Shapiro Wilk test for the $(\Sigma_1, \Sigma_2)$ interval fails. When extending the interval to $(-0.16m, 0.16m)$ – which covers the contributions of the peak assignment mode well –, the direct statistical analysis for the 348 points estimates a standard deviation of $0.056$m, skew of $0.06$ and a kurtosis of $0.21$. This suggest a symmetrical distribution that is slightly higher with wider wings, which is consistent with the above observation of the refinement mode being recognizable in interval and the standard deviation is overestimated. A Shapiro Wilk test supports normal distribution in the interval $(-0.16m, 0.16m)$, as $W = 0.995$ and the corresponding p-Value p=0.38. As we already excluded outliers for the analysis, there is no meaning in the statistical analysis of the refinement distance and we assume $\Sigma$ as a measure for the systematic deviation when using cubic spline interpolation.

The cubic spline scheme confirms the above observed maximal error of the match points $\Delta D = 0.043$m $= \sigma_A \approx \sigma_E = 0.044$. When restricting $|\delta|, |\Sigma| \leq 0.375$m, then both $\delta$ and $\Sigma$ have 3 match points exceeding this threshold and both distributions fit a profile with a standard deviation of $\sigma_F = \sigma_B$ (without 3 outliers) $= 0.13$m for the refinement each match point contributes in average. This means that one of the interpolation schemes is not superior to the other, but comparing them illustrates the uncertainty associated with interpolation in between the match points.

Now, we calculate the systematic deviation between linear and cubic spline interpolation from datasets in 0.01 m resolution. $\zeta_i$ denotes the maximal absolute difference in the interval between the i-th and (i+1)-th match point, which is a direct measure of systematic differences due to the interpolation schemes.

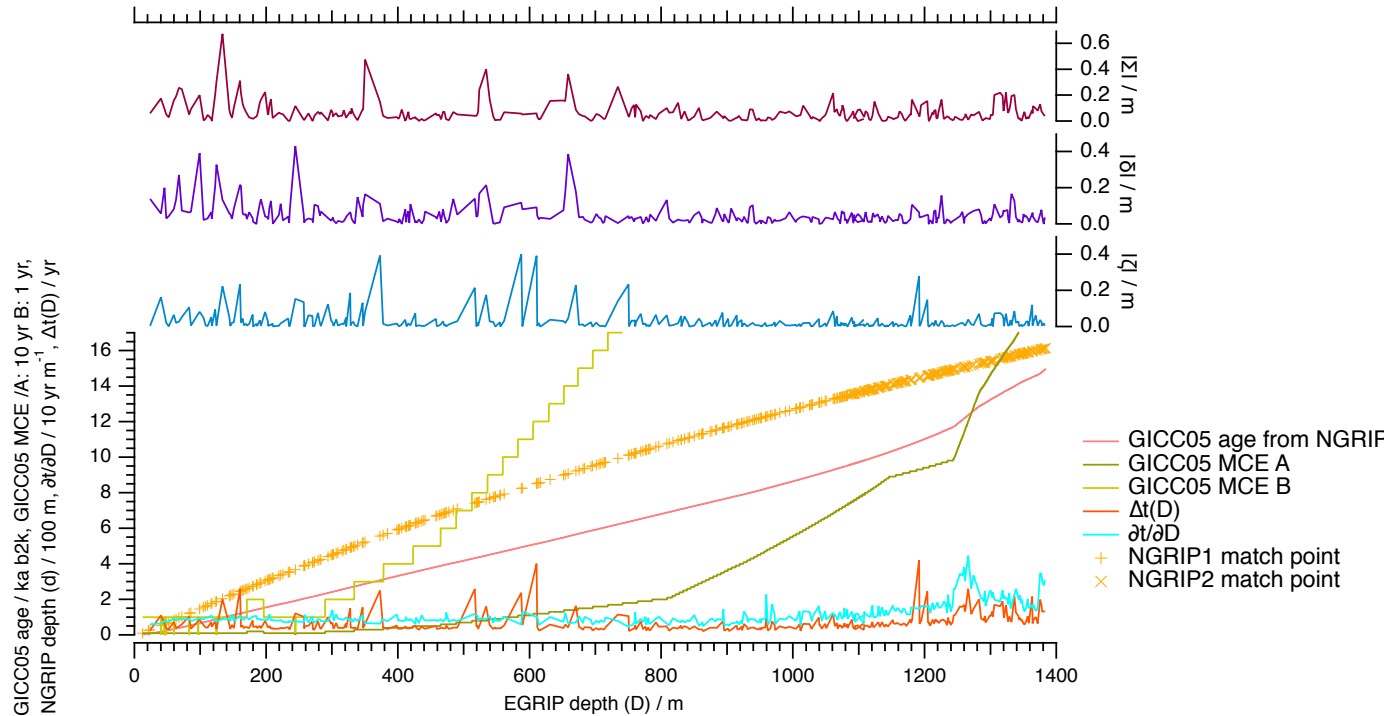

**Figure E2.** The (EGRIP, NGRIP) match points, (EGRIP-depth, GICC05-time)-scale, the (EGRIP-depth, GICC05-MCE)-curve, $|\delta|$, $|\Sigma|$, $|\zeta|$, $\frac{\partial t_{\text{GICC05}}}{\partial D_{\text{EGRIP}}}$, $\Delta t(D) = \left| \frac{\partial t}{\partial D} \right| \sqrt{(\Delta t)^2 + \zeta^2}$

$|\zeta|$ and $|\delta|$ are both less than 0.4 m and exhibit a similar pattern, while $|\Sigma|$ has less in common with both $\delta$ and $\zeta$. $|\zeta|$ is a
good measure for the interpolation uncertainty along the record as it is the direct comparison of two fundamentally different interpolation approaches (see Fig. E2).

For linear interpolation, the statistical error for the computed depth in between two match points is limited by the maximal error of the match points $\Delta D = 0.043\text{m}$ and the error of the interpolated depth D is therefore $\sqrt{(\Delta D)^2 + \zeta^2}$. To propagate the depth error and estimate the additional error of the time match, we start from the highest resolution published GICC05 dating
of NGRIP with 2.5 cm and 5 cm depth resolution above and below 349.8 m respectively (Vinther et al., 2006; Rasmussen et al., 2006) and linearly interpolate the EGRIP depth ($D$) onto the NGRIP depth ($d$) to get the time scale t(D) for EGRIP. We calculate $\frac{\partial t_{\text{GICC05}}}{\partial D_{\text{EGRIP}}}$ in the high resolution dataset and sample it at the match points.

The matching error related to the timescale transfer $\Delta t(D) = \left| \frac{\partial t}{\partial D} \right| \sqrt{(\Delta t)^2 + \zeta^2}$ is maximally about 4 years, exceeds the MCE on two occasions in the uppermost 200 m by 1 year, and becomes increasingly smaller compared to the MCE for
increasingly deeper parts of the record (see Fig. E2).

*Author contributions.* Original draft preparation by SM with major contributions from FW, SOR, EC, MSJ; statistical analysis of the (depth,depth)-match and interpolation by FW and SOR; matching by SM, FW, SOR, GS, MSJ; DEP data processing by SM, FW; ECM data processing by SOR, GS, MSJ; tephra data processing by EC, SD, GJ; DEP measurements in the field by SM, SK, NAB, SHF, VG, HK; ECM measurements in the field by SM, SOR, HK, TE, KN, SMPB, SHF, IK; preparation, set-up and testing of DEP system by SM, FW; preparation, set-up and testing of ECM system by BV, DDJ, SOR; tephra selection and sampling in the field by EC, SM, SD, SMPB; All authors contributed to improving the final paper.

*Acknowledgements.* We thank Frédéric Parrenin and 2 anonymous reviewers for fruitful comments that improved the paper a lot. We thank all people involved in logistics, drilling and ice-core processing in the field. EGRIP is directed and organized by the Centre for Ice and Climate at the Niels Bohr Institute, University of Copenhagen. It is supported by funding agencies and institutions in Denmark (A. P. Møller Foundation, University of Copenhagen), USA (US National Science Foundation, Office of Polar Programs), Germany (Alfred Wegener Institute, Helmholtz Centre for Polar and Marine Research), Japan (National Institute of Polar Research and Arctic Challenge for Sustainability), Norway (University of Bergen and Bergen Research Foundation), Switzerland (Swiss National Science Foundation), France (French Polar Institute Paul–Emile Victor, Institute for Geosciences and Environmental research) and China (Chinese Academy of Sciences and Beijing Normal University). Sune Olander Rasmussen and Giulia Sinnl gratefully acknowledge the Carlsberg Foundation for support to the project ChronoClimate. Sergio Henrique Faria acknowledges support from the project iMechPro (RTI2018–100696–B–I00) from the Spanish Ministry of Science, Innovation, the Spanish Government through the María de Maeztu excellence accreditation 2018–2022 (Ref. MDM–2017–0714) and the Basque Government through the BERC 2018–2021 programme. Tobias Erhardt acknowledges the long-term support of ice-core research by the Swiss National Science foundation (SNFS) and the Oeschger Center for Climate Change Research.

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
