# Peer review of "A first chronology for the East GReenland Ice-core Project (EGRIP) over the Holocene and last glacial termination"

_Climate of the Past, 2019_

## Referee Comment (RC1) · Anonymous Referee #1 · 11 Dec 2019

General Comments:

This paper presents an initial timescale for the EGRIP ice core from Greenland. Overall, this manuscript is relatively straightforward and the scientific context and results are presented well. However, I have two primary concerns that I think need to be addressed before this manuscript is suitable for final publication.

The topic of this paper is transferring the existing GICC05 timescale to the new EGRIP core through volcanic tie points. While ample detail is provided on the existing ice cores and on measurement details, very little explanation is given to the details of linking GICC05 with the 373 tie points. The entirety of this process, is briefly summarized

in a short paragraph in section 2.4 whereby a simple linear interpolation is used to link these tie points and the uncertainty associated with GICC05 is transferred to GICC05-EGRIP-1, with little to no treatment of the additional uncertainties associated with the transfer. I think the methods the authors used may be fine, but they definitely need much more explanation, clarification and justification than is offered here. Please see the comment below for Line 225, since I also think that some further analysis is needed both in justifying the interpolation scheme as well as in quantifying the timescale uncertainty.

There are issues with writing clarity and grammar with some mistakes (which I have not completely listed). These issues make it difficult to completely understand the manuscript. While the quality of writing is high enough to understand most of the science presented in this manuscript, I would recommend that the authors spend time refining the grammar and sentence structure of the paper to improve readability.

While these issues are important, I believe that they can be addressed by the authors in a revised version of this manuscript. The research presented so far clearly represents a lot of work and it is exciting to see new progress from the EGRIP project. Thank you for your efforts so far!

Specific Comments:

Line 6: Are the 373 match points spaced throughout the entire ice core?

Line 7-8: How deep is the core in total? Do you have a total age estimate?

Line 15: change 'reflect' to 'reflecting'. Not sure what 'immediate' means in this context.

Figure 1: Excellent figure.

Line 78: Change "was" to "were".

Lines 78-94: Are these procedures novel and unique to this study? If so, I would recommend including a diagram or schematic. If very similar methods have been using

[Figure]

previously, referencing them in this section would be helpful.

Figure 3: I would suggest writing out in plain language the y-axis label and including units in the axis labels and/or caption.

Lines 113-114: It would be interesting to know some information about the amount of breaks or missing ice at various depths. I would suggest adding a few descriptive statistics on core quality at different depth and especially in the brittle zone.

Lines 119-120: It is unclear to me what 'it' or 'protocol' are referring to in this sentence.

Lines 123-124: This sentence needs more context. Why does the DEP data need temperature correction? How did you accomplish this and at what stage in your procedure? I (and most readers) have not been to EGRIP, so we will need some explanation of what the 'science trench' and 'core buffer' mean and their implications for the DEP data.

Figure 4: What percentage of the data was removed? For permittivity it looks like the 'bad quality' measurements encompass a large amount of data.

Line 143: I suggest switching 'used' to 'final'.

Line 150-151: I am having difficulty following this section since 'bag marks' and 'break marks' have not been clearly defined.

Figure 6: Why not show Mazama data from NGRIP? In any case, this is a very convincing figure.

Section 2.4: Did you set quantitative thresholds for how much accumulation variability and core smoothness, or were the results inspected qualitatively. If the former, what were the assumptions you used?

Figure 8: There should be only two Es in NEEM on the y-axis label. Also what do the pink/red bars in the brittle ice zone signify?

Section 3.2: If I understand correctly, you found 3 matching tephra horizons out of 373 total matches. Is this correct? How many other tephra events have been sampled so far? How many more do you plan to sample? Have there been many other events sampled that do not match any event in NGRIP or NEEM? This continuous tephra sampling is very impressive and interesting and more details would be appreciated.

Line 221: 1383.84 meters in EGRIP right?

Line 225: What is the longest section between tie points? 0-2 years seems an unrealistically low uncertainty to report if there is no annual layer counting. We can see in Figure 9 that accumulation rate changes on multiple timescales and presumably has variations within the spacing of your tie points as well. None of the purely mathematical interpolation methods will account for this possibility. I think you need to include some analysis that incorporates the observed variation in annual layer thickness, either from layers visible in the EGRIP ice core or from meteorological data. You can use this data in conjunction with your tie points and their spacing to generate more realistic estimates of uncertainty and potentially improve the timescale itself.

Line 226-230: I'm afraid I do not fully understand either of these sentences, which I think are important. I would suggest adding clarification.

Line 236: How do you know the upstream accumulation is higher? Is there a reference for this? Or are you inferring this from the flatness of the 0-8 ka accumulation curve in Fig. 9? How do you separate the spatial versus temporal signal in reconstructed accumulation?

Line 239-240: The phrase "EGRIP layers start to get thinner, but remain nearly constant in thickness" seems to be a direct contradiction. Please clarify.

Line 249: How deep is the full core and what is its anticipated age?

Line 260: Why not upload the timescale also at annual resolution to be more useful for other users? I'm assuming that you will include match point data for all 373 matches

as well at the 3 tephra horizons reported here.

---

## Referee Comment (RC2) · Anonymous Referee #2 · 6 Jan 2020

Mojtabavi et al. present a first chronology for the EastGRIP ice core. The chronology extends to 15ka and is based on volcanic events identified with electrical measurements as well as three tephra layers. The work provides many useful methods and the chronology appears accurate; however, the paper is sloppily written. This is a methods heavy paper despite the technologies being mature. There is little other analysis in the manuscript beyond basic plots of annual layer thickness.

An initial timescale is a useful result. The authors describe the electrical measurements well, something that has not been done as part of recent papers. This may be because the methods have changed little from the papers of the 80s and 90s that describe

them. Regardless, I found the discussions mostly useful. Timescale papers are often a challenge to write because they take lots of work, but little in the way of direct scientific conclusions come of the timescale alone – the papers that measure specific climate parameters which depend on the timescale get the high profile results. Thus, the lack of new insight is not necessarily a negative. However, I kept asking myself if the work presented is sufficient for a stand-alone publication, because in many ways it feels like an interim timescale – useful, but not needed to be published.

I reviewed previous ice-core timescale papers published in Climate of the Past to better understand the contribution of this manuscript relative to its peers. I compared to Winski t al. (2019) for SPICEcore, Ramussen et al. (2013) for NEEM, and Sigl et al. (2016) and Buizert et al. (2015) for WAIS Divide. Each of these papers is quite different than this work. However, what stands out is that the larger scope of each of these manuscripts. Winski et al. is most comparable, being an ice-phase timescale only; however, it presents a timescale for the full ice core (55 ka), includes chemical in addition to electrical measurements, and provides annual layer interpolation for the Holocene although it does not provide any tephra analysis. Sigl et al. is also an ice-phase timescale only, but presents 31ka of annual layers that provides a reference chronology for all of Antarctica, if not all ice cores – it is clearly in a different class of both effort and impact. The Rasmussen et al. paper is more comparable being the most recent timescale published for a Greenlandic core. It develops an ice-phase timescale with similar methods to this manuscript, but does so for entirety of the non-folded core and also includes a gas timescale. There is a considerable analysis of the accumulation rate history from the core as well. Buizert et al. determine the gas timescale (and ice timescale for ages older than 31ka) and, because of the low delta-age, this work is a major improvement for Antarctic timescales. Thus, Mojtabavi et al. are considerably short of the benchmark set by previous timescale papers in Climate of the Past.

I am not sure, however, that a comparison to other papers is the appropriate metric.

Or rather, I think the editor is better suited to make this call. I think my role is better limited to whether what is presented in the paper is useful. In this case, my question becomes: is this publication useful beyond what simply publishing the timescale and electrical data sets would be. And I believe it is, even if by only a little. There are 10 figures, but most are rather simple. Despite this, I found the text interesting and the detail useful, despite the sloppy writing.

Making electrical measurements of ∼1400m of ice is time consuming and both the timescale and electrical measurements are undeniably useful. I wish the authors had described why they decided to truncate the timescale at 15ka. More EastGRIP ice has been drilled and more is left to be drilled. Is there a scientific reason for this? Or is it simply a logistical one – like a grant running out? I provide additional comments below and will leave the decision of whether this manuscript achieves the standards of Climate of the Past to the editor. But I will add that if the authors wish to add analysis, comparing the EastGRIP timescale to that predicted by the traced radar layers (Vallelonga et al., 2014; Christianson et al., 2014) used in site selection would be interesting and useful.

Detailed comments:

Introduction – the introduction is lacking a review of relevant literature and how timescales are being developed and used. With the exception of two Joughin references for NEGIS, every reference is for GICC05. This is not a full introduction. This should be a subsection on GICC05, with an actual introduction that is much broader in scope and discusses the work beyond this group's narrow niche. The lack of any GISP2 references regarding annual layer interpretation is also notable.

L11-12: I don't understand what this last sentence is trying to say

L14: The first sentence needs to be reworded. The grammar is not correct.

L15: What is the point of the second sentence? Which core? Why 25,000 years

L23 – be specific about what EastGRIP will tell us about ice dynamics at the onset of NEGIS

L23 – be specific about the ages being investigated. "half way through the glacial period" is very vague

L28 – given that you stop this timescale at 15ka, is how the GICC05 is built beyond 14.7ka really very important?

L30: Sentence starting "The time scale" says nothing.

L32 – "back" is written twice

L36 – I don't think chemo-stratigraphic is a good abbreviation/conglomeration. Just say how they were synchronized

L38 – Use active voice

L40 – why only this time range? Ice from deeper has already been collected, and presumably measured

L42: NEEM should be defined on first usage. And do you ever define NGRIP?

L51: change "around" to "about". Around has spatial connotations which makes it confusing to use here.

2.1.2 – Isn't this just a repeat of the introduction?

L55 – why mention older parts of the GICC05 timescale – you are only going back to 15ka

L70 – "Dielectric profiling (DEP) has been introduced as a system for rapid dielectrical profiling...". The sloppiness of the writing needs to be addressed.

L71 – delete "recorded"

L78 – change "was" to "Were"

L79 – you don't need "respectively"

L80 – provide the typical values so that the you can illustrate the variations are indeed slight

L82 – what offset?

L83-85 – this needs to be reworded for clarity

L85-85 – if these variations are slight, why spend so much time writing about them? Is this necessary?

L95-105 – This paragraph needs to be reworked. There are a lot of details here, many of which aren't actually needed for this work and that don't seem to have a real point. Yes, the free air capacitance matters for some things, but lay this out logically. And if the reader doesn't need to know this for this work, say it in a sentence rather than a paragraph.

L101 – there is no reason to cite an in prep paper here. You can just say that this is something that is or could be done in the future.

L109 – what is the unit here?

L132 – how much data total was not collected?

L138 – I don't think you really mean "calibrated". I think "adjusted" would be a fairer description

L156-160 – Why do this conversion if you are only going to say it's not correct and not important? Just stop the bad practice of reporting this as acidity, which is known to be wrong. Readers from outside our field may not quickly notice that the acidity is known to be not accurate.

L180-193. I understand what they authors are trying to get at here, but the wording is difficult to follow. This section needs to be rewritten.

L195 – what is a "slight annual layer thickness variation"?

L197 – This paragraph needs to be rewritten as well. I don't understand how the depths and ages are being transferred among cores. Why is there an interpolation for each ice core bag? If you are interpolating between match points, is which bag the ice is part of irrelevant?

L200 – where do you discuss uncertainties based on linear and cubic spline interpolations?

L207 – what was the decision process for which tie points were kept and which were not?

L209-211 – This is not a sentence

L226 – I don't understand what you are trying to say here about getting ages with linear interpolation of EGRIP? How different? What is unrealistic? What is the correlation of recent annual accumulation rates between NGRIP and EGRIP? Is there a justification that the annual variability in NGRIP is appropriate to map to EGRIP? Why are you referencing Rasmussen et al. 2013?

L229 – rewrite this sentence

L236 – "supposedly" Cite a source for this information and write with precision.

Figure 9: The large changes in annual layer thickness variation over short periods are a bit concerning in some locations. Most notably, at ∼13ka, where the annual layer thickness varies by a factor of 3 or so from 0.03m to 0.09m. This looks to me like a bad match in between two good matches. The authors need to more rigorously assess instances of abrupt layer thickness change.

---

## Author Comment (AC1) · 3 Jul 2020

Please see the response letters in attach pdf file.

Please also note the supplement to this comment:
https://cp.copernicus.org/preprints/cp-2019-143/cp-2019-143-AC1-supplement.pdf

---

## Author Comment (AC2) · 3 Jul 2020

**Answers to Reviewer 1**

Dear Referee 1,

Thank you very much for the constructive comments. We followed your suggestions and have extended and restructured the concerned sections. For clarity we add our comments into your review just below the respective paragraphs.

General Comments: This paper presents an initial timescale for the EGRIP ice core from Greenland. Overall, this manuscript is relatively straightforward and the scientific context and results are presented well. However, I have two primary concerns that I think need to be addressed before this manuscript is suitable for final publication.

(1) comments from Referees: The topic of this paper is transferring the existing GICC05 timescale to the new EGRIP core through volcanic tie points. While ample detail is provided on the existing ice cores and on measurement details, very little explanation is given to the details of linking GICC05 with the 373 tie points.

**(1r) author's response:** We will detail the description of how the match points themselves were assessed in the section "Synchronization of dielectric profiling and electrical conductivity measurement records of EGRIP, NGRIP & NEEM" and add a paragraph discussing the distance of match points at its end:

**(1c) author's changes:** To provide in depth information, we added detailed information in subsection 2.4 "Synchronization of dielectric profiling and electrical conductivity measurement records of EGRIP, NGRIP & NEEM". Now, we elaborate on how the match was actually done in much more detail and the initial paragraph reads now: "Patterns in the DEP records of NGRIP, NEEM and EGRIP were initially matched by one investigator. The same cores' ECM data were matched separately and independently by three different investigators. Both matches are mainly based on clearly identifiable volcanic peaks and also synchronous patterns of other events (Figure 7 in the revised manuscript), which not necessarily need to be of volcanic origin, but are assumed to reflect synchronous events. Based on these independent matches, the four investigators identified consistent and reliable common patterns, that are represented in the ECM and/or the DEP records from NGRIP and at least one of the other ice cores. For the confirmation of match points, all records of all three cores were loaded into the Matchmaker tool [Rasmussen et al., 2013] and assessed jointly by all four investigators in the different display options featured by the software. The Matchmaker tool allows easy identification of wrong match points via interactive plots and on-line evaluation of the match. To validate match points, we plot the depths of the common match points $D_i$ (in EGRIP or NEEM) against $d_i$ (NGRIP). The slope of each of these (depth, depth)-curves is the annual layer thickness ratio of the two cores, $r_i = \frac{D_{i+1} - D_i}{d_{i+1} - d_i}$. Points which deviate from the (depth, depth) curve or create jumps in $r$, are easily recognized and checked again. We only expect significant abrupt changes in $r$ at times where the climate (and thus the relative accumulation rates) shifts due to changes in climate conditions [Rasmussen et al., 2006, Rasmussen et al., 2013, Seierstad et al., 2014, Winski et al., 2019], while the different ice-flow patterns at the cores' sites only lead to slow changes in $r$. Short-term accumulation variability due to both climatic factors and wind-driven redistribution of snow on the surface can lead to relatively large variations in the ratio of layer thicknesses between different cores, especially when match points are only a few years apart. To reduce short-term accumulation-rate variability in the final timescale, we re-evaluated intervals with large variability in annual-layer-thickness ratios, and removed too closely spaced match points. The final minimum distance between match points is 0.22 m (1206.45m-1206.67m), corresponding to around 3 years. Overall, the match points are reasonably evenly distributed throughout the entire ice core, and the maximum distance

between neighbouring match points is 26.6 m (490.06 m – 516.67 m), corresponding to a time interval of 224 years."

(2) comments from Referees: The entirety of this process, is briefly summarized in a short paragraph in section 2.4 whereby a simple linear interpolation is used to link these tie points and the uncertainty associated with GICC05 is transferred to GICC05- EGRIP-1, with little to no treatment of the additional uncertainties associated with the transfer. I think the methods the authors used may be fine, but they definitely need much more explanation, clarification and justification than is offered here.

**(2r) author's response:** The procedure of transferring the timescale by tie points is discussed in detail in Rasmussen et al. (2013; paragraph 3.2) for the transfer from NGRIP to NEEM. The transfer from NGRIP to EGRIP is entirely the same procedure with the same inferred uncertainties and errors. Rasmussen et al. (2013) discuss the additional uncertainty of the transfer from NGRIP GICC05modelext to NEEM: "The accuracy of the timescale at these points depends on three factors:

1. The NEEM timescale inherits the maximum counting error (MCE) of the NGRIP GICC05 timescale.

2. Differences between the shape of peaks and inaccuracies in the depth registration of the ECM data set introduce synchronization uncertainty on the order of centimetres. The estimated synchronization uncertainty was estimated to 10 cm ($1\sigma$) by [Rasmussen et al., 2008], and here we tentatively estimate its magnitude by calculating the effect on the (NEEM depth, NGRIP depth) relation of removing every second match point. The results support the estimated synchronization uncertainty of 10 cm ($1\sigma$), leading to timescale transfer uncertainties ranging from a few years to a maximum of a few decades at the deepest part of the record.

3. Although we believe the set of match points to be robust, there is a risk that some sections have been erroneously matched up, leading to a larger systematic depth offset.

As the MCE is typically 2 orders of magnitude larger than the matching uncertainty (when assuming no large systematic errors), we report GICC05modelext-NEEM-1 ages with the MCE uncertainty estimates only, but stress that observed phasing differences of up to a decade at the match-point depths could be artefacts from the timescale transfer."

**(2c) author's changes:** We extended the discussion in the section 2.5 and subsection 2.5.1:

[revised manuscript text omitted]

(3) comments from Referees: Please see the comment below for Line 225, since I also think that some further analysis is needed both in justifying the interpolation scheme as well as in quantifying the timescale uncertainty.

**(3r) author's response:** This is discussed with the response to referee comment for line 225.

**(3c) author's changes:** The changes are described with the response to referee comment for line 225.

(4) comments from Referees: There are issues with writing clarity and grammar with some mistakes (which I have not completely listed). These issues make it difficult to completely understand the

manuscript. While the quality of writing is high enough to understand most of the science presented in this manuscript, I would recommend that the authors spend time refining the grammar and sentence structure of the paper to improve readability. While these issues are important, I believe that they can be addressed by the authors in a revised version of this manuscript. The research presented so far clearly represents a lot of work and it is exciting to see new progress from the EGRIP project. Thank you for your efforts so far!

**(4r) author's response:** We acknowledge that this has to be done!

**(4c) author's changes:** We carefully edited the manuscript and made some grammatical changes (refer to the synopsis of the revised and the original manuscript please). Below we address all the individual comments.

(5) comments from Referees:Specific Comments: Line 6: Are the 373 match points spaced throughout the entire ice core?

**(5r) author's response:** The match points are reasonably distributed throughout the entire ice core. The maximum distance to the neighbouring match points is 26.6 m (490.06m-516.67m), corresponding to around 224 yr difference and the minimum distance is 0.21m between match points (1206.45m-1206.67m), corresponding to around 3 years. Note: When compiling the datasets for release, we realized a small inconsistency, which we corrected by adding 8 more match points over the brittle zone 373 to 381). We revised our figures and the manuscript accordingly. We also chose to add a short not on why we kept a certain minimum distance in between match points to reduce noise on e.g. the annual layer thickness record.

**(5c) author's changes:** We changed the number of match points and added "typically spaced less than 50 years apart" in the abstract, as the reviewer seems to want know if the spacing of the match points is reasonable. The sentence reads now: "We transfer the annual-layer-counted Greenland Ice Core Chronology 2005 (GICC05) from the NGRIP core to the EGRIP ice core by means of 381 match points, typically spaced less than 50 years apart."

To provide in depth information, we added the detailed information in subsection 2.4 "Synchronization of dielectric profiling and electrical conductivity measurement records of EGRIP, NGRIP & NEEM": "Short-term accumulation variability due to both climatic factors and wind-driven redistribution of snow on the surface can lead to relatively large variations in the ratio of layer thicknesses between different cores, especially when match points are only a few years apart. To reduce short-term accumulation-rate variability in the final timescale, we re-evaluated intervals with large variability in annual-layer-thickness ratios, and removed too closely spaced match points. The final minimum distance between match points is 0.22 m (1206.45m-1206.67m), corresponding to around 3 years. Overall, the match points are reasonably evenly distributed throughout the entire ice core, and the maximum distance between neighbouring match points is 26.6 m (490.06 m – 516.67 m), corresponding to a time interval of 224 years."

(6) comments from Referees: Line 7-8: How deep is the core in total? Do you have a total age estimate?

**(6r) author's response:** As the abstract is limited in length, we add this information in the data and methods section.

**(6c) author's changes:** We added the following half-sentence to section 2.2.1: "The average annual accumulation rate is about 100 kg m-2 yr-1 (0.11 m i.e. yr-1) for the period 1607–2011 as determined from a firn core close to the main EGRIP drilling site [Vallelonga et al., 2014]. Radar-soundings suggest the ice thickness to exceed 2550 m and traced radar layers from the NGRIP site suggest that the drill site preserves an undisturbed climatic record of at least 51 kyr [Vallelonga et al., 2014]. The camp currently moves about 51 m to the North-Northeast each year [Dahl-Jensen et al., 2019]."

(7) comments from Referees: Line 15: change 'reflect' to 'reflecting'. Not sure what 'immediate'

means in this context.

**(7r) author's response:** We added "that" and omitted "immediate", which was meant to strengthen the close relation of the proxies to the atmospheric proxies or even direct measurement of greenhouse gases.

**(7c) author's changes:** The sentence reads now: "The dating of an ice core establishes the depth–age relationship to derive a chronology of past climatic conditions from the measured proxy parameters, which reflect past atmospheric conditions and biogeochemical events along the core."

(8) comments from Referees: Figure 1: Excellent figure.

**(8r) author's response:** Thank you very much!

(9) comments from Referees: Line 78: Change "was" to "were".

**(9c) author's changes:** Changed.

(10) comments from Referees: Lines 78-94: Are these procedures novel and unique to this study? If so, I would recommend including a diagram or schematic. If very similar methods have been using previously, referencing them in this section would be helpful.

**(10r) author's response:** Similar methods have not been described previously, so it is not done with just referencing them. The system is – in principle – the one as described in [Wilhelms et al., 1998]. The paragraph you refer to, describes improvements and typical measurements with the device.

**(10cc) author's changes:** We added the following schematic Figure 4 to the 2.3.1 section to illustrate the description in the text to be clear for the non-expert readers.

[Figure]

Figure 4: Schematic of the DEP instrument.

(11) comments from Referees: Figure 3: I would suggest writing out in plain language the y-axis label and including units in the axis labels and/or caption.

**(11c) author's changes:** Changed as suggested (Figure 5).

[Figure]

Figure 5: Calibration curve of the measurements with artificial tubes and free-air measurement for DEP device.

(12) comments from Referees: Lines 113-114: It would be interesting to know some information about the amount of breaks or missing ice at various depths. I would suggest adding a few descriptive statistics on core quality at different depth and especially in the brittle zone.

**(12c) author's changes:**

We added the following information to the 2.2.1 section. "The EGRIP brittle zone is of better quality than the brittle ice from previous Greenland ice core projects such as NEEM and NGRIP. For the EGRIP core, Figure 6 presents a quality index on the basis of the ratio between validated and total measured DEP and ECM sample points. This quality index falls below 0.3 between 505 m (4220 a b2k) and 1210 m (11163 a b2k) depth, which is consistent with the brittle zone between 550 m and 1250 m according to the field season reports Dahl2019. The quality index calculated from the earlier released NGRIP and NEEM DEP data is presented in Appendix A "Quality index for the NGRIP and NEEM ice cores."

[Figure]

Figure 6: Match points between EGRIP, NEEM (blue) and NGRIP (yellow) ice cores based on the DEP and ECM data sets. The core quality index Q as derived from the validated DEP and ECM data, respectively.

We have added a section for the NGRIP and NEEM ice cores quality in the Appendix (A):

**Appendix A: Quality index for the NGRIP and NEEM ice cores**

For the NEEM and NGRIP ice cores we calculated similar quality indices as provided for EGRIP above. They are presented together in Figure 7.

[Figure]

Figure 7: Quality indices for the EGRIP, NGRIP and NEEM ice cores.

(13) comments from Referees: Lines 119-120: It is unclear to me what 'it' or 'protocol' are referring to in this sentence.

**(13r) author's response:** Initially, the "gold standard" DEP protocol was to note all breaks, classify bad core quality sections with e.g. missing slices and core catcher marks in a hand written protocol book in a code that is easily processed into a record of validated/non validated core sections. For the three deep cores the processing staff did not stick strictly to the protocol format in a consistent way. On the other hand, when really locking at the records the fraction of rejected core sections is pretty big and a significant amount of data can be validated after observing a consistent and intact permittivity signal. This procedure can be automated by calculating a strongly smoothed average of the permittivity, setting a threshold and removing sections that fall below the threshold. To make the edges a little nicer the removed section is extended to the depth where it reaches the average before resp. after dropping below resp. exceeding the threshold. This was described in section 2.3. of [Rasmussen et al., 2013]. The conductivity is less sensitive to bad core quality, which leads to a really robustly validated conductivity record, when taking the decision on the permittivity record. Finally, this automated procedure delivers a comparable result, with much less effort, and most important: it is much more consistent in between all three cores.

**(13c) author's changes:** We changed the section 2.3.1 and focused on the actually used procedure. It reads now: "The automated procedure as described in [Rasmussen et al., 2013] (section 2.3) is much faster, more consistent in between the three different cores, and has proven to be superior to any approach based on a hand-written protocol, which depends on the judgement of the operator when identifying intervals of bad core. As the permittivity is very sensitive to bad core quality and

the conductivity is much less prone to bad core quality, the outlined validation procedure leads to a robustly validated conductivity record."

(14) comments from Referees: Lines 123-124: This sentence needs more context. Why does the DEP data need temperature correction? How did you accomplish this and at what stage in your procedure? I (and most readers) have not been to EGRIP, so we will need some explanation of what the 'science trench' and 'core buffer' mean and their implications for the DEP data.

**(14r) author's response:** In contrast to the permittivity, the conductivity is strongly dependent on temperature. The temperature correction does not affect the identification of conductivity patterns of DEP peaks between synchronised ice cores. Therefore it is not relevant for the synchronisation in this paper. However, it is relevant when using absolute conductivity readings, especially with respect to the attenuation of radar waves in the ice. As people will use the dataset for this purpose, a warning has to be placed somewhere in the paper. The processing mode changes in between projects and teams. The temperature is difficult to reconstruct later as many people are involved and the measurement of the temperature of the core when processing is difficult as the equilibration time between moving it from the relatively cold storage area (buffer) to the science trench with much more impact from the surface air temperature as more people move in and out may vary in between teams. For the cores here we can't reliably reconstruct the precise temperature during the measurement, so we will not provide temperature corrected data.

**(14c) author's changes:** We replaced the sentence, it reads now: "Due to the varying temperature in processing area throughout the field seasons, the core was not processed at a consistent temperature and we don't have the temperature readings avail to provide consistently harmonized conductivity data. The missing temperature correction does not affect the use of conductivity peaks for synchronization purposes in between ice cores, which is relevant for the discussion here. When e.g. deriving radar wave absorption coefficients from the presented conductivity record, one would have to be very cautious and have this limitation of the data in mind."

(15) comments from Referees: Figure 4: What percentage of the data was removed? For permittivity it looks like the 'bad quality' measurements encompass a large amount of data.

**(15r) author's response:** As we discussed in the first version of manuscript (2.2.1 section, lines 115-125), the ice cores exhibit breaks, broken-off slices are clearly identifiable in the permittivity record by dropping spikes. For the validation of the data, any drop below a certain threshold (cf. the red line in Figure 4 in the revised manuscript) identifies a spike to be rejected, where the segment to be rejected is extended to about the average of the permittivity record. In this way, with an average with 11% data was not validated in the entire ice core (especially in the brittle zone). It is not bad quality measurements. The measurement is recorded along all the core, even on "damaged" sections, which can even be identified in the permittivity record and thus be removed from the validated dataset.

**(15c) author's changes:** A core quality index was introduced under item 12, which answers the question.

(16) comments from Referees: Line 143: I suggest switching 'used' to 'final'.

**(16c) author's changes:** Changed as suggested.

(17) comments from Referees: Line 150-151: I am having difficulty following this section since 'bag marks' and 'break marks' have not been clearly defined.

**(17r) author's response:** Each 1.65 m ice core section contains the equivalent of three 0.55 m bags, and the position of the bag marks separating the first and second bags and the second and third bags, respectively. The break mark is referring to the any break on the ice core sections.

**(17c) author's changes:** Introduced the definition of break marks somewhat further up. The section reads now: "Also, the core-break positions were registered along with measurement by

moving the electrodes of the ECM instrument to the respective break position after the core scan, and registering the position in the data file. During the processing, these recorded break marks were used to trim off artefacts and produce the final ECM data set. Data from each day were calibrated using independent measurements of the physical dimensions of the ECM measurement setup. The first and last few millimetres of recorded data are affected by the proximity to the end of the core and were removed. Areas with dips in the signal around logged core breaks were also muted during processing. Details on the acquisition and processing of the ECM record are laid out in Appendix C "Details on the ECM procedures."

(18) comments from Referees: Figure 6: Why not show Mazama data from NGRIP? In any case, this is a very convincing figure.

**(18r) author's response:** The MAZAMA data on the figure is not from NGRIP, because it is unpublished geochemical data from Siwan Davies. The statistics in the table (SD and D2 are however based on the NGRIP data). Although she was happy for the depth to be used, she wants to publish the data in a specific tephra paper, which is in preparation, first.

(19) comments from Referees: Section 2.4: Did you set quantitative thresholds for how much accumulation variability and core smoothness, or were the results inspected qualitatively. If the former, what were the assumptions you used?

**(19r) author's response:** The accumulation variability was inspected qualitatively. The match is described in "Synchronization of dielectric profiling and electrical conductivity measurement records of EGRIP, NGRIP & NEEM", also the chosen minimal distance of the match points. Distance. In response to reviewer remark (2), the errors of the new timescale are discussed in depth in the added discussion to this section 2.5.

**(19c) author's changes:** The discussion is addressed in handling of reviewer comment (23) and (2).

(20) comments from Referees: Figure 8: There should be only two Es in NEEM on the y-axis label. Also what do the pink/red bars in the brittle ice zone signify?

**(20r) author's response:** In item 12 we introduced a quality index which replaces the brittle zone indication.

**(20c) author's changes:** We have fixed the y-axis label.

(21) comments from Referees: Section 3.2: If I understand correctly, you found 3 matching tephra horizons out of 373 total matches. Is this correct? How many other tephra events have been sampled so far? How many more do you plan to sample? Have there been many other events sampled that do not match any event in NGRIP or NEEM? This continuous tephra sampling is very impressive and interesting and more details would be appreciated.

**(21r) author's response:** So far, only the DEP /ECM peaks were investigated for tephra in EGRIP, as the project is still in an early stage for tephra screening. This targeted sampling led to the identification of 13 tephra layers in total, three of which link to NGRIP. The other layers have not yet been linked geochemically to other ice cores. They will be traced in NEEM and NGRIP as a next step. Eliza Cook took continuous samples for tephra in the field, with an 11 cm depth resolution, and these will be mounted onto slides and investigated by optical microscopy over the next few years, due to the high number of samples. It has been shown [Davies et al., 2010] that tephra layers are often found without coeval chemical peaks in DEP or ECM, so we anticipate many more 'cryptotephra' ash deposits to be found. Furthermore, there is no detailed tephra framework published yet for the Holocene. Eliza Cook and Siwan Davies are working on continuous sampling for tephra in NGRIP and, with some samples for NEEM and GRIP a. The former has been continuously sampled between 5 ka to the beginning of the Holocene (and back to the glacial), and the other cores have only been spot sampled in the Holocene, to trace the most interesting layers found in NGRIP. These

remain unpublished but will undoubtedly lead to many more correlations between all the ice cores and will be published at a later date.

**(21c) author's changes:** n.a.

(22) comments from Referees: Line 221: 1383.84 meters in EGRIP right?

**(22r) author's response:** Yes. 1383.84 meters in EGRIP.

(23) comments from Referees: Line 225: What is the longest section between tie points? 0-2 years seems an unrealistically low uncertainty to report if there is no annual layer counting. We can see in Figure 9 that accumulation rate changes on multiple timescales and presumably has variations within the spacing of your tie points as well. None of the purely mathematical interpolation methods will account for this possibility. I think you need to include some analysis that incorporates the observed variation in annual layer thickness, either from layers visible in the EGRIP ice core or from meteorological data. You can use this data in conjunction with your tie points and their spacing to generate more realistic estimates of uncertainty and potentially improve the timescale itself.

**(23r) author's response:** As we have discussed before in Line 6, "The final minimum distance between match points is 0.22 m (1206.45m-1206.67m), corresponding to around 3 years. Overall, the match points are reasonably evenly distributed throughout the entire ice core, and the maximum distance between neighbouring match points is 26.6 m (490.06 m – 516.67 m), corresponding to a time interval of 224 years." Figure 9 shows the annual layer thickness between the match points. Here we used the ages of the match points themselves against annual layer thicknesses between them to make our plot (not interpolation data). We added the detailed information of interpolations in subsection 2.5 and two Appendix sections (D and E).

**(23c) author's changes:** Regarding uncertainty of interpolations please see changes (2c). At the end of the section "Synchronization of dielectric profiling and electrical conductivity measurement records of EGRIP, NGRIP & NEEM" we added a paragraph discussing the distance of match points: "Short-term accumulation variability due to both climatic factors and wind-driven redistribution of snow on the surface can lead to relatively large variations in the ratio of layer thicknesses between different cores, especially when match points are only a few years apart. To reduce short-term accumulation-rate variability in the final timescale, we re-evaluated intervals with large variability in annual-layer-thickness ratios, and removed too closely spaced match points. The final minimum distance between match points is 0.22 m (1206.45m-1206.67m), corresponding to around 3 years. Overall, the match points are reasonably evenly distributed throughout the entire ice core, and the maximum distance between neighbouring match points is 26.6 m (490.06 m – 516.67 m), corresponding to a time interval of 224 years."

(24) comments from Referees: Line 226-230: I'm afraid I do not fully understand either of these sentences, which I think are important. I would suggest adding clarification.

**(24c) author's changes:** We changed the paragraph, it reads now: "Along with this publication we release a time scale for each 0.55 m section ("bag"). For each EGRIP depth, the corresponding NGRIP depth was found by linear interpolation between the match points, and the GICC05 age was then determined from the published GICC05 time scale for NGRIP. The maximal uncertainty resulting from the choice of interpolation scheme is assessed in detail (see Appendix E1) and is about four years. The relatively smooth (depth, depth) relation of EGRIP–NGRIP and EGRIP–NEEM (see Figure 6) shows that the ratios of annual layer thicknesses between cores do not vary noticeably between match points. Figure 8 shows that EGRIP has thinner annual layers than both NEEM and NGRIP ice cores in the upper parts of the cores as also expected from the lower surface accumulation. Ice found in the EGRIP core originates from snow that was accumulating upstream, and accumulation rates increase upstream as the flow line approaches GRIP and NGRIP, where present-day accumulation is about twice of that at EGRIP tc-8-1275-2014,Riverman,karlsson2020.

Surprisingly, annual layers in EGRIP remain almost constant back to 8 ka b2k (Figure 9 in the revised manuscript), while the layer thicknesses in large parts of the Holocene part of the NGRIP and NEEM cores thin linearly due to ice flow. We believe that it is a coincidence that the combined effects of the increasing upstream accumulation and flow-induced thinning at EGRIP balance out for the last 8 ka. Despite the lower accumulation at EGRIP, annual layers in EGRIP eventually get thicker than the annual layers in the NEEM and NGRIP ice cores. Below an EGRIP depth of around 700 m, annual layers in EGRIP are thicker than the layers from the same period in the NEEM core, and similarly below 1000 m, EGRIP annual layers are thicker than those in NGRIP (Figure 8).There are some gaps in the EGRIP ice-core record due to the brittle zone. However, the smoothness of the depth vs. depth plot in Figure 6 and the annual layer thickness ratio in Figure 8 robustly support our time scale based on the match points."

[Figure]

Figure 8: The EGRIP/NGRIP (orange) and EGRIP/NEEM (blue) annual-layer thickness ratio (left axis) calculated between neighbouring match points.

(25) comments from Referees: Line 236: How do you know the upstream accumulation is higher? Is there a reference for this? Or are you inferring this from the flatness of the 0-8 ka accumulation curve in Fig. 9? How do you separate the spatial versus temporal signal in reconstructed accumulation?
**(25r) author's response:** Has been added in the paragraph as discussed in Reviewers remark (24).
**(25c) author's changes:** See no. (24) above.
(26) comments from Referees: Line 239-240: The phrase "EGRIP layers start to get thinner, but

remain nearly constant in thickness" seems to be a direct contradiction. Please clarify

**(26r) author's response:** This was clarified in the changes made for the reviewer remarks above, esp. (24).

(27) comments from Referees: Line 249: How deep is the full core and what is its anticipated age?

**(27r) author's response:** This information was added to section EGRIP: Radar-soundings suggest the ice thickness to exceed 2550 m and traced radar layers from the NGRIP site suggest that the drill site preserves an undisturbed climatic record of at least 51 kyr [Vallelonga et al., 2014].

**(27c) author's changes:** n.a.

(28) comments from Referees: Line 260: Why not upload the timescale also at annual resolution to be more useful for other users? I'm assuming that you will include match point data for all 373 matches as well at the 3 tephra horizons reported here.

**(28r) author's response:** The annually resolved time scale will mainly be useful when the data upon which it is based are also available. The full-resolution NGRIP CFA data set will be released in connection with another paper which is currently in review, and the plan is to release the GICC05 annual layer count in the same context.

**(28c) author's changes:** n.a.

**Answers to Reviewer 2**

Dear Referee 2,

Thank you very much for reviewing our manuscript and for your constructive comments. We sincerely appreciate the time you have spent on our manuscript. Please find the detailed point-by-point response to your comments below.

(29) comments from Referees: Mojtabavi et al. present a first chronology for the EastGRIP ice core. The chronology extends to 15ka and is based on volcanic events identified with electrical measurements as well as three tephra layers. The work provides many useful methods and the chronology appears accurate; however, the paper is sloppily written. This is a methods heavy paper despite the technologies being mature. There is little other analysis in the manuscript beyond basic plots of annual layer thickness.

An initial timescale is a useful result. The authors describe the electrical measurements well, something that has not been done as part of recent papers. This may be because the methods have changed little from the papers of the 80s and 90s that describe them. Regardless, I found the discussions mostly useful. Timescale papers are often a challenge to write because they take lots of work, but little in the way of direct scientific conclusions come of the timescale alone – the papers that measure specific climate parameters which depend on the timescale get the high profile results. Thus, the lack of new insight is not necessarily a negative. However, I kept asking myself if the work presented is sufficient for a stand-alone publication, because in many ways it feels like an interim timescale – useful, but not needed to be published.

**(29r) author's response:** It is an interim timescale until an annual-layer-counted timescale will be become available (possibly a multi-core annual-layer counted time scale updating/replacing GICC05), but it will likely take several (if not many) years before such a time scale becomes available for the past 15 ka, so in that context, we believe that having a time scale to work with in the meantime is valuable. The delay of this year's field season due to the Corona virus outbreak even prolongs the time span till the whole core is available. The timescale will be the basis for several topical papers focussing on shorter time intervals, as e.g. the Holocene, younger Holocene

or transition and different proxies. From that point of view it makes sense to publish one proper treatment of the timescale and not repeat it in several papers that need the timescale as a fundamental basis for their discussions. The detailed descriptions of the DEP system and associated procedures comprise the evolution and improvements of the system during the past two decades compared to [Wilhelms et al., 1998] and have not been described in detail before, while they are relevant for the treatment of our data.

**(29c) author's changes:** n.a.

(30) comments from Referees: I reviewed previous ice-core timescale papers published in Climate of the Past to better understand the contribution of this manuscript relative to its peers. I compared to Winski t al. (2019) for SPICEcore, Ramussen et al. (2013) for NEEM, and Sigl et al. (2016) and Buizert et al. (2015) for WAIS Divide. Each of these papers is quite different than this work. However, what stands out is that the larger scope of each of these manuscripts. Winski et al. is most comparable, being an ice-phase timescale only; however, it presents a timescale for the full ice core (55 ka), includes chemical in addition to electrical measurements, and provides annual layer interpolation for the Holocene although it does not provide any tephra analysis. Sigl et al. is also an icephase timescale only, but presents 31ka of annual layers that provides a reference chronology for all of Antarctica, if not all ice cores – it is clearly in a different class of both effort and impact. The Rasmussen et al. paper is more comparable being the most recent timescale published for a Greenlandic core. It develops an ice-phase timescale with similar methods to this manuscript, but does so for entirety of the nonfolded core and also includes a gas timescale. There is a considerable analysis of the accumulation rate history from the core as well. Buizert et al. determine the gas timescale (and ice timescale for ages older than 31ka) and, because of the low deltaage, this work is a major improvement for Antarctic timescales. Thus, Mojtabavi et al. are considerably short of the benchmark set by previous timescale papers in Climate of the Past.

I am not sure, however, that a comparison to other papers is the appropriate metric. Or rather, I think the editor is better suited to make this call. I think my role is better limited to whether what is presented in the paper is useful. In this case, my question becomes: is this publication useful beyond what simply publishing the timescale and electrical data sets would be. And I believe it is, even if by only a little. There are 10 figures, but most are rather simple. Despite this, I found the text interesting and the detail useful, despite the sloppy writing. Making electrical measurements of 1400m of ice is time consuming and both the timescale and electrical measurements are undeniably useful. I wish the authors had described why they decided to truncate the timescale at 15ka. More EastGRIP ice is it simply a logistical one – like a grant running out? I provide additional comments below and will leave the decision of whether this manuscript achieves the standards of Climate of the Past to the editor. But I will add that if the authors wish to add analysis, comparing the EastGRIP timescale to that predicted by the traced radar layers (Vallelonga et al., 2014; Christianson et al., 2014) used in site selection would be interesting and useful.

**(30r) author's response:** The manuscript presents new ECM and DEP data (and a thorough account of new DEP processing) and these data sets will be released with the paper (down to 1383.84 m). In addition, for the first time we will release the DEP data from NGRIP1 (down to 1298.705 m) and NEEM (down to 1493.297 m) ice cores, and publishing this manuscript will open these data sets to analysis by the community.

- GICC05-EGRIP-1 time scale for the EGRIP ice core

- Specific conductivity measured with the dielectric profiling (DEP) technique on the EGRIP ice core, 13.77-1383.84 m depth

- Permittivity measured with the dielectric profiling (DEP) technique on the EGRIP ice core, 13.77-1383.84 m depth

- Concentration of hydrogen ions measured with the Electrical Conductivity Method (ECM) on the EGRIP ice core (down to 1383.84 m depth)

- Specific conductivity measured with the dielectric profiling (DEP) technique on the NEEM ice core (down to 1493.297 m depth)

- Permittivity measured with the dielectric profiling (DEP) technique on the NEEM ice core (down to 1493.297 m depth)

- Specific conductivity measured with the dielectric profiling (DEP) technique on the NGRIP1 ice core (down to 1372 m depth)

- Permittivity measured with the dielectric profiling (DEP) technique on the NGRIP1 ice core (down to 1372 m depth)

The current data reach into MIS 3, and even if the current drilling and processing plans hold, it will take at least two years before the complete EGRIP ECM/DEP data sets will be ready. Regarding your question of why we stop specifically at 15 ka: The density of the match points in the Holocene, GS-1 (often called the Younger Dryas) and GI-1 (or the Bølling-Allerød) is so much higher than in the glacial (and especially in the LGM) that the nature of the time scale transfer procedure is different, making it a natural point to cut. Also, ice-flow modelling will be more important to understand the deeper than the upper half of the core, and this work is not ready yet. Therefore, this study goes back through the Holocene and last glacial termination. As your suggestion to compare our results with the traced radar layers [Vallelonga et al., 2014, Christianson et al., 2014]. We have found out that the resolution and depth precision of the traced radar layers is insufficient to really be useful.

**(30c) author's changes:** The above explanation why only going back 15 ka, because of match point density we add as authors changes for the following detailed comment (31c).

(31) comments from Referees: Detailed comments: Introduction – the introduction is lacking a review of relevant literature and how timescales are being developed and used. With the exception of two Joughin references for NEGIS, every reference is for GICC05. This is not a full introduction. This should be a subsection on GICC05, with an actual introduction that is much broader in scope and discusses the work beyond this group's narrow niche. The lack of any GISP2 references regarding annual layer interpretation is also notable.

**(31r) author's response:** Annual layer counting is not within the scope of this paper. Annually counted records are also compared and accessed on the basis of identifiable match points. The GISP2 project was implemented more than 25 years ago and the main analysis was published almost 2 decades ago, to significant extent in the AGU special issue "Greenland Summit Ice Cores" in 1997, where annual layer counting of the GISP2 core was presented. We have cited the [Vinther et al., 2006, Rasmussen et al., 2006],paper, where GRIP, NGRIP and DYE-3 were re-counted resp. counted further back in time within the effort to create the GICC05 timescale. There is detailed discussion of the findings for GICC05 in comparison to the GISP2 results, by means of comparison at well identifiable match points for a similar time period to the one covered here. This was 14 years ago and since GISP2 only the group around the University of CPH has been active in acquiring ice core records beyond few thousand years in Greenland. Where several members of the GISP2 consortium are now partners within the efforts we undertake in Greenland.

**(31c) author's changes:** We added a subsection for the GICC05:

**2.1 GICC05**

"The annual-layer-counted Greenland Ice Core Chronology 2005 (GICC05) is derived from measurements of stable water isotopes in the DYE–3, GRIP and NGRIP ( see Figure 1 in the revised manuscript) ice cores for the period back to 7.9 ka b2k [Vinther et al., 2006] and high-resolution measurements of chemical impurities, conductivity of the ice, and visual stratigraphy from the GRIP and NGRIP ice cores for the period between 7.9 ka and 14.7 ka b2k [Rasmussen et al., 2006]. For the period from 14.7 ka to 42 ka b2k, the dating of the cores is based on annual layer counting in the visual stratigraphy, the electrical conductivity profiles, and a set of chemical impurities data [Andersen et al., 2006]. The timescales are compared to time scales of different other climate archives at suitable tie points, like e.g. marine sediment cores [Svensson et al., 2006]. For the NGRIP core, the GICC05 time scale has been extended even further into the glacial, back to 60 ka b2k by annual layer counting [Svensson et al., 2008] and ice-flow modelling [Wolff et al., 2010]. For the older parts [Wolff et al., 2010] the NGRIP ss09sea06bm model time scale, shifted to younger ages by 705 years, has been spliced onto the end of the GICC05 timescale, thereby forming the so-called GICC05modelext chronology. The GICC05modelext was also applied to the central Greenland GRIP and GISP2 cores by more than 900 marker points and verification with 24 tephra horizons [Seierstad et al., 2014]. In summary, the GICC05modelext timescale is the consistent reference frame for the entirety of Greenland deep cores."

(32) comments from Referees: L11-12: I don't understand what this last sentence is trying to say

**(32c) author's changes:** The sentence is changed to: "For the next years, this initial timescale will be the basis for climatic reconstructions from EGRIP high-resolution proxy data sets, like e.g. stable water isotopes, chemical impurity or dust records."

(33) comments from Referees: L14: The first sentence needs to be reworded. The grammar is not correct.

**(33c) author's changes:** The sentence reads now: "The dating of an ice core establishes the depth–age relationship to derive a chronology of past climatic conditions from the measured proxy parameters, which reflect past atmospheric conditions and biogeochemical events along the core."

(34) comments from Referees: L15: What is the point of the second sentence? Which core? Why 25,000 years

**(34c) author's changes:** We have removed this sentence as we found this is not really essential to mention here.

(35) comments from Referees: L23 – be specific about what EastGRIP will tell us about ice dynamics at the onset of NEGIS

**(35c) author's changes:** We added the following sentence to the introduction section: "A main objective of the EGRIP project is to study the dynamics of the ice flow in the NEGIS ice stream by analysing the ice core's rheology and its relation to the deformation of the ice."

(36) comments from Referees: L23 – be specific about the ages being investigated. "half way through the glacial period" is very vague

**(36c) author's response:** The remark was relating to the original EGRIP proposal. We clarified the 15 ka, this paper is focussing on and relate to the present interglacial and the transition from the last glacial in the same paragraph.

**(36c) author's changes:** We have removed the remark.

(37) comments from Referees: L28 – given that you stop this timescale at 15ka, is how the GICC05 is built beyond 14.7ka really very important?

**(37r) author's response:** The remark addresses the introduction section. Few lines are spent to outline how the GICC05modelext timescale was established and as we match till 14.96 kyr b2k, there are 3 references out of the mentioned 4 that address the sections of GICC05 that are used for the match here. On the other hand a good match below also supports the match above.

**(37c) author's changes:** n.a.

(38) comments from Referees: L30: Sentence starting "The time scale" says nothing.

**(38r) author's response:** This was a fraction that was left behind during an edit and indeed says nothing.

**(38c) author's changes:** We reworded together with section addressed in 37r above: "For the period from 14.7 ka to 42 ka b2k, the dating of the cores is based on annual layer counting in the visual stratigraphy, the electrical conductivity profiles, and a set of chemical impurities data [Andersen et al., 2006]. The timescales are compared to time scales of different other climate archives at suitable tie points, like e.g. marine sediment cores [Svensson et al., 2006]."

(39) comments from Referees: L32 – "back" is written twice

**(39r) author's response:** We deleted the first occurrence.

**(39c) author's changes:** The sentence is changed to: "For the NGRIP core, the GICC05 time scale has been extended even further into the glacial, back to 60 ka b2k by annual layer counting [Svensson et al., 2008] and ice-flow modelling [Wolff et al., 2010]."

(40) comments from Referees: L36 – I don't think chemo-stratigraphic is a good abbreviation/conglomeration. Just say how they were synchronized

**(40r) author's response:** The term chemo-stratigraphic is taken from [Seierstad et al., 2014] and is common in the literature to describe chemical peak matching. Whether to hyphenate, may depend on journal style.

**(40c) author's changes:** n.a.

(41) comments from Referees: L38 – Use active voice

**(41c) author's changes:** The sentence is changed to: "To apply this approach to the EGRIP core, we have profiled the upper 1383.84 m of the EGRIP core using ECM and DEP in the field during the 2017, 2018 and 2019 field seasons."

(42) comments from Referees: L40 – why only this time range? Ice from deeper has already been collected, and presumably measured

**(42r) author's response:** See changes (31c) and (34c).

**(42c) author's changes:** n.a.

(43) comments from Referees: L42: NEEM should be defined on first usage. And do you ever define NGRIP?

**(43r) author's response:** self-evident, of course.

**(43c) author's changes:** We fixed it in the introduction, in the abstract there are still abbreviations. If it should be fixed in the abstract, please tell us.

(44) comments from Referees: L51: change "around" to "about". Around has spatial connotations which makes it confusing to use here.

**(44c) author's changes:** Changed.

(45) comments from Referees:2.1.2 – Isn't this just a repeat of the introduction?

**(45r) author's response:** We introduced a GICC05 paragraph in the data and methods section and moved most of the detailed discussion there, where it appears along with the paragraphs on EGRIP, NGRIP and NEEM.

(46) comments from Referees: L55 – why mention older parts of the GICC05 timescale – you are only going back to 15ka

**(46r) author's response:** see comments (31c) and (37c).

(47) comments from Referees: L70 – "Dielectric profiling (DEP) has been introduced as a system for rapid dielectrical profiling: : :". The sloppiness of the writing needs to be addressed.

**(47r) author's response:** Dielectric profiling (DEP) is a proper noun for a method/device and rapid dielectric profiling is what it is used for. So we believe that the sentence makes sense.

**(47c) author's changes:** We also added the schematic Figure 4 to the 2.3.1 section as suggested by reviewer 1, and added more description. Please see comment (10c).

(48) comments from Referees: L71 – delete "recorded"

**(48c) author's changes:** Done.

(49) comments from Referees: L78 – change "was" to "Were"

**(49c) author's changes:** Changed.

(50) comments from Referees: L79 – you don't need "respectively"

**(50c) author's changes:** Done.

(51) comments from Referees: L80 – provide the typical values so that the you can illustrate the variations are indeed Slight

**(51r) author's response:** Pure glacier ice's conductivity is in the order of 15µS/m. The conductance $G = \sigma C0/\varepsilon 0 = 15 S/m * 63 fF/(8.854 pF/m) = 110 nS$. The ice permittivity is $\geq 2$ and the capacitance therefore $C \geq 125 fF$. (Figure 4 in the revised manuscript). But it is more obvious, when converting the given values to comparable changes of the material properties, the reader can check himself against the baseline in Figure 4 in the revised manuscript.

**(51c) author's changes:** We added the following sentence to the Appendix B: Calibration and corrections to the DEP data: "The slight capacitance and conductance variation on the order of less than 4 fF and 500 pS, thus corresponding to relative permittivity changes of 4 fF/63 fF = 0.06 and conductivity changes of (500 pS)/(63 fF)*(8.8542 pF) = 70 nS, along the DEP device is due to the unavoidable deformation of the cables (Figure 4a) when moving the scanning electrode along the device (Figure 4e)." Compared to the properties of pure glacier ice (ref. to Fig. 4) these variations are in the order of 2% for the permittivity and 5‰ for the conductivity. Additionally, an offset of few nS residual conductance may remain even after performing the correction routines of the LCR meter (inductance L, capacitance C, resistance R) bridge (Figure 4c)."

(52) comments from Referees: L82 – what offset?

**(52r) author's response:** The residual conductance variation due to the deformation of the cables.

**(52c) author's changes:** We reworded for clarity and the sentence reads now: "Additionally, an offset of few nS residual conductance may remain even after performing the correction routines of the LCR meter (inductance L, capacitance C, resistance R) bridge (Figure 4c)."

(53) comments from Referees: L83-85 – this needs to be reworded for clarity

**(53r) author's response:** We added Figure 4 that presents a cable carrier that is deformed during movement.

**(53c) author's changes:** We added the following sentence to the Appendix B: Calibration and corrections to the DEP data: "As a correction in the few percent range we correct the offset, introduced by the changing stray admittance due to the varying cable geometry (Figure 4a) due to their movement during the measurement, by subtracting the course of free-air measurements from the respective measurement of a core section along the DEP device when processing the data."

(54) comments from Referees: L85-85 – if these variations are slight, why spend so much time writing about them? Is this necessary?

**(54r) author's response:** We addressed the variations in reviewer comment 51 and demonstrate they are about 2% and 5‰. When using the data for e.g. synthetic radargrams the errors are in a range to be considered.

**(54c) author's changes:** n.a.

(55) comments from Referees: L95-105 – This paragraph needs to be reworked. There are a lot of details here, many of which aren't actually needed for this work and that don't seem to have a real point. Yes, the free air capacitance matters for some things, but lay this out logically. And if the reader doesn't need to know this for this work, say it in a sentence rather than a paragraph.

**(55r) author's response:** We restructured the whole DEP section and the information of this paragraph is distributed and put where being in a better context. The assessment of procedures and errors is mainly relevant to further use of the data, which is also discussed in the text. It is not relevant to that extent for the synchronisation of the cores. We moved this part to Appendix B "Calibration and corrections to the DEP data".

(56) comments from Referees: L101 – there is no reason to cite an in prep paper here. You can just say that this is something that is or could be done in the future.

**(56c) author's changes:** We have removed this prep citation.

(57) comments from Referees: L109 – what is the unit here?

**(57r) author's response:** Relative permittivity is dimensionless.

**(57c) author's changes:** We clarified by adding "relative" permittivity in Figure 5:"the effective relative permittivity of the setup"

(58) comments from Referees: L132 – how much data total was not collected?

**(58r) author's response:** Around 8.5 m was not collected between 1295 to 1395!

**(58c) author's changes:** We added the information by extending a sentence to: "The section about 1285–1385m was corrected in this way, where in total about 8.5 m of the 100 m were not measured. Furthermore, we relied more heavily on the ECM record than on the DEP record when matching peaks within sections with known problems."

(59) comments from Referees: L138 – I don't think you really mean "calibrated". I think "adjusted" would be a fairer Description

**(59c) author's changes:** Changed as suggested.

(60) comments from Referees: L156-160 – Why do this conversion if you are only going to say it's not correct and not important? Just stop the bad practice of reporting this as acidity, which is known to be wrong. Readers from outside our field may not quickly notice that the acidity is known to be not accurate.

**(60r) author's response:** The conversion is not accurate, even an inaccurate acidity conversion is more physical meaningful than the measured current. As long as we give the details of the conversion and a better conversion is not available, we argue that this is a meaningful way to show the data

**(60c) author's changes:** n.a.

(61) comments from Referees: L180-193. I understand what they authors are trying to get at here, but the wording is difficult to follow. This section needs to be rewritten.

**(61c) author's changes:** The changes are described with the response to referee 1 comment. Please see comment (1c).

(62) comments from Referees: L195 – what is a "slight annual layer thickness variation"?

**(62r) author's response:** We assume similar annual layer thickness variability between EGRIP and NGRIP, corresponding to an only slowly varying slope of the NGRIP vs EGRIP match-point depth curve (see Figure 6). We expected only significant change in annual layer thickness at the two drill sites at climatic transitions and thinning factor due to ice flow only changes slowly with depth. Rasmussen et al. (2013) used the same assumption and approach for interpolating between NEEM and NGRIP match points.

**(62c) author's changes:** n.a.

(63) comments from Referees: L197 – This paragraph needs to be rewritten as well. I don't understand how the depths and ages are being transferred among cores. Why is there an interpolation for each ice core bag? If you are interpolating between match points, is which bag the ice is part of irrelevant?

**(63r) author's response:** The entire paragraph has been re-written when addressing reviewer comment (2) of reviewer 1.

**(63c) author's changes:** please see comment (2c).

(64) comments from Referees: L200 – where do you discuss uncertainties based on linear and cubic spline interpolations?

**(64r) author's response:** The changes are described with the response to referee 1 comment. Please see comment (2c).

**(64c) author's changes:** please see comment (2c).

(65) comments from Referees: L207 – what was the decision process for which tie points were kept and which were not?

**(65r) author's response:** this was already discussed, when addressing reviewer 1 comment (23).

**(65c) author's changes:** At the end of the section "Synchronization of dielectric profiling and electrical conductivity measurement records of EGRIP, NGRIP & NEEM" we added a paragraph discussing the distance of match points: "Short-term accumulation variability due to both climatic factors and wind-driven redistribution of snow on the surface can lead to relatively large variations in the ratio of layer thicknesses between different cores, especially when match points are only a few years apart. To reduce short-term accumulation-rate variability in the final timescale, we re-evaluated intervals with large variability in annual-layer-thickness ratios, and removed too closely spaced match points. The final minimum distance between match points is 0.22 m (1206.45m-1206.67m), corresponding to around 3 years. Overall, the match points are reasonably evenly distributed throughout the entire ice core, and the maximum distance between neighbouring match points is 26.6 m (490.06 m – 516.67 m), corresponding to a time interval of 224 years."

(66) comments from Referees: L209-211 – This is not a sentence

**(66c) author's changes:** The sentence reads now: "The ECM and DEP do not follow each other closely in the 1245–1283 m interval because of the alkaline nature of the ice associated with stadial conditions in EGRIP. This is due to high dust levels neutralizing the acidity of the ice [Ruth et al., 2003, Rasmussen et al., 2013]."

(67) comments from Referees: L226 – I don't understand what you are trying to say here about getting ages with linear interpolation of EGRIP? How different? What is unrealistic? What is the correlation of recent annual accumulation rates between NGRIP and EGRIP? Is there a justification that the annual variability in NGRIP is appropriate to map to EGRIP? Why are you referencing Rasmussen et al. 2013?

**(67r) author's response:** We rewrote and updated the paragraph.

**(67c) author's changes:** We changed the paragraph, it reads now: "Along with this publication we release a time scale for each 0.55 m section ("bag"). For each EGRIP depth, the corresponding NGRIP depth was found by linear interpolation between the match points, and the GICC05 age was then determined from the published GICC05 time scale for NGRIP. The maximal uncertainty resulting from the choice of interpolation scheme is assessed in detail (see Appendix E1) and is about four years. The relatively smooth (depth, depth) relation of EGRIP–NGRIP and EGRIP–NEEM (see Figure 6) shows that the ratios of annual layer thicknesses between cores do not vary noticeably between match points. Figure 8 shows that EGRIP has thinner annual layers than both NEEM and NGRIP ice cores in the upper parts of the cores as also expected from the lower surface accumulation. Ice found in the EGRIP core originates from snow that was accumulating upstream, and accumulation rates increase upstream as the flow line approaches GRIP and NGRIP, where

present-day accumulation is about twice of that at EGRIP tc-8-1275-2014,Riverman,karlsson2020. Surprisingly, annual layers in EGRIP remain almost constant back to 8 ka b2k (Figure 9 in the revised manuscript), while the layer thicknesses in large parts of the Holocene part of the NGRIP and NEEM cores thin linearly due to ice flow. We believe that it is a coincidence that the combined effects of the increasing upstream accumulation and flow-induced thinning at EGRIP balance out for the last 8 ka. Despite the lower accumulation at EGRIP, annual layers in EGRIP eventually get thicker than the annual layers in the NEEM and NGRIP ice cores. Below an EGRIP depth of around 700 m, annual layers in EGRIP are thicker than the layers from the same period in the NEEM core, and similarly below 1000 m, EGRIP annual layers are thicker than those in NGRIP (Figure 8).There are some gaps in the EGRIP ice-core record due to the brittle zone. However, the smoothness of the depth vs. depth plot in Figure 6 and the annual layer thickness ratio in Figure 8 robustly support our time scale based on the match points."

(68) comments from Referees: L229 – rewrite this sentence

**(68c) author's changes:** The sentence reads now: "The relatively smooth (depth, depth) relation of EGRIP–NGRIP and EGRIP–NEEM (see Figure 6) shows that the ratios of annual layer thicknesses between cores do not vary noticeably between match points."

(69) comments from Referees: L236 – "supposedly" Cite a source for this information and write with precision.

**(69r) author's response:** The references Vallelonga et al. (2014), Riverman et al. (2019) and Karlsson et al. (2020) were added to the sentence before starting from L235 and we corrected the grammar.

**(69c) author's changes:** Added a further sentence: "Ice found in the EGRIP core originates from snow that was accumulating upstream, and accumulation rates increase upstream as the flow line approaches GRIP and NGRIP, where present-day accumulation is about twice of that at EGRIP tc-8-1275-2014,Riverman,karlsson2020."

(70) comments from Referees:Figure 9: The large changes in annual layer thickness variation over short periods are a bit concerning in some locations. Most notably, at 13ka, where the annual layer thickness varies by a factor of 3 or so from 0.03m to 0.09m. This looks to me like a bad match in between two good matches. The authors need to more rigorously assess instances of abrupt layer thickness change.

**(70r) author's response:** We have fixed the issue. Two match points were very close to each other that caused this large change. We added further description on the distance of match points.

**(70c) author's changes:** " Short-term accumulation variability due to both climatic factors and wind-driven redistribution of snow on the surface can lead to relatively large variations in the ratio of layer thicknesses between different cores, especially when match points are only a few years apart. To reduce short-term accumulation-rate variability in the final timescale, we re-evaluated intervals with large variability in annual-layer-thickness ratios, and removed too closely spaced match points. The final minimum distance between match points is 0.22 m (1206.45m-1206.67m), corresponding to around 3 years. Overall, the match points are reasonably evenly distributed throughout the entire ice core, and the maximum distance between neighbouring match points is 26.6 m (490.06 m – 516.67 m), corresponding to a time interval of 224 years."

Note: The language of the manuscript was polished.

[revised manuscript text omitted]

---

## Author Response (AR2)

**Dear Editor and Reviewers,**

Thank you very much for fruitful comments that improved the paper a lot. We sincerely appreciate the time the reviewers have spent on the manuscript. For clarity we add our comments into your review just below the respective paragraphs.

**Answers to Dr. Frédéric Parrenin,**

Comments from Referee: This paper aims at producing a first chronology for the EGRIP ice core in Greenland. EGRIP is synchronized onto NGRIP and NEEM by the mean of volcanic matching (381 match points), and the GICC05 agescale is transferred onto EGRIP. The match is checked at 3 points where tephras allow for an unambiguous match. The new agescale is named GICC05-EGRIP-1 and extend back to 15 kyr b2k.

This is a simple and straightforward, yet useful paper. It went already through a round a review, thus it is quite polished already. I only one main comment regarding the interpolation in-between the match points and the error analysis. If I understood the manuscript correctly, this error analysis is done in an absolute way, although I would expect the interpolation error to be linked to the distance to the nearest tie point. Such analysis would provide an added value for the manuscript. We focused our analysis on the match points themselves, but we also provided the discussion you suggest en passent. The interpolation error is - in general - very hard to assess, as it strongly depends on the used interpolation scheme. When assuming linear interpolation the error is less than the maximal error of the adjacent match points we interpolate in between.

To assess this problem we discuss in section 2.5.1: "However, when we want to know the age at an arbitrary EGRIP depth, additional uncertainties apply due to the interpolation between the match points. There are two dominant sources: As discussed above, variations in relative accumulation rates and ice flow may add up to a handful of years of additional uncertainty relative to GICC05, but there is also a contribution from the choice of interpolation scheme in between the match points. The difference introduced by the choice between the most widely used linear and cubic spline interpolation schemes (Press et al., 1992) is about an order of magnitude larger than the above-mentioned random uncertainty associated with the identification of the match points (see Appendix E)."

In the last 3 paragraphs of "Appendix E: Statistical analysis of the (depth, depth)-match with cubic spline interpolation" we assess it: "Now, we calculate the systematic deviation between linear and cubic spline interpolation from datasets in 0.01 m resolution.  $\zeta_i$  denotes the maximal absolute difference in the interval between the i-th and (i+1)-th match point, which is a direct measure of systematic differences due to the interpolation schemes."

 $|\zeta|$  and  $|\delta|$  are both less than 0.4 m and exhibit a similar pattern, while  $|\Sigma|$  has less in common with both  $\delta$  and  $\zeta$ .  $|\zeta|$  is a good measure for the interpolation uncertainty along the record as it is the direct comparison of two fundamentally different interpolation approaches (see Figure E2).

For linear interpolation, the statistical error for the computed depth in between two match points is limited by the maximal error of the match points  $\Delta D = 0.043$ m and the error of the interpolated depth D is therefore  $\sqrt{(\Delta D)^2 + \zeta^2}$ . To propagate the depth error and estimate the additional error of the time match, we start from the highest resolution published GICC05 dating of NGRIP with 2.5 cm and 5 cm depth resolution above and below 349.8 m respectively (Vinther et al., 2006; Rasmussenet al., 2006) and linearly interpolate the EGRIP depth (D) onto the NGRIP depth (d) to get the time scale t(D) for EGRIP. We calculate  $\frac{\partial t_{GICC05}}{\partial D_{EGRIP}}$  in the high resolution dataset and sample it at the match points.

The matching error related to the timescale transfer  $\Delta t(D) = \left|\frac{\partial t}{\partial D}\right| \sqrt{(\Delta t)^2 + \zeta^2}$  is maximally about 4 years, exceeds the MCE on two occasions in the uppermost 200 m by 1 year, and becomes increasingly smaller compared to the MCE for increasingly deeper parts of the record (see Figure E2)."

**Technical comments:**

1. 15-16: first sentence is heavy.

We have split the paragraph in two sentences, it reads now: "The dating of an ice core establishes the depth–age relationship to derive a chronology of past climatic conditions from the measured proxy parameters. The proxy parameters reflect past atmospheric conditions and biogeochemical events along the core."

1. 57: please use superscripts for m-2 and yr-1.

Changed.

1. 58: please format citation.

Corrected.

1. 63: figure 8 does not seem to be numbered by the order of appearance. Corrected.

**Answers to Reviewer 2**

Comments from Referee: The writing has been improved from the original submission as well as the methods being more fully described. The additional details on the measurements in the appendices is also quite useful and a significant improvement. Minor comments are in the attached pdf as sticky notes.

The paper does still lack much in the way of scientific discussion. To address this, I recommend that the authors add a discussion of the accuracy of the existing Holocene and late glacial chronologies. Specifically, the GISP2 chronology is in better agreement with IntCal13 at 10 and 11 ka when GICC05 was shown diverge (Adolphi and Muscheler, 2016; Rasmussen et al., 2006). Therefore, it is not clear GICC05 is the best choice of timescale to tie the EastGRIP core to during this time period. This information was not available when the NEEM timescale was developed, so will be useful to revisit the accuracy of the annually resolved Greenland ice-core reference chronologies in the Holocene and late Glacial. Such a discussion would not only be useful to readers of this paper, but it could also help motivate analysis of EastGRIP to specific time periods where the chronologies could be improved with new annual layer interpretations.

We added the paragraph to the end of the 2.1 GICC05 section: "Previous studies assessed the differences between independent timescales of Holocene paleoclimate records. Adolphi and Muscheler (2016) indicated that the GICC05 counting error underestimates the total uncertainty in some parts of the Holocene based on the comparison between the radiocarbon dating calibration curve (Reimer et al., 2013, IntCal13) and (Svensson et al., 2008, GICC05), and the work was extended in Adolphi et al. (2018). The objective of this work, however, is to extend GICC05 to the EGRIP core to allow parallel analysis of the records, and we thus refrain from a further discussion of the absolute accuracy of GICC05 here."

**A first chronology for the East GReenland Ice–core Project (EGRIP) over the Holocene and last glacial termination**

Seyedhamidreza Mojtabavi1,2, Frank Wilhelms1,2, Eliza Cook3, Siwan Davies4, Giulia Sinnl3, Mathias Skov Jensen3, Dorthe Dahl-Jensen3,5, Anders Svensson3, Bo Vinther3, Sepp Kipfstuhl1,3, Gwydion Jones4, Nanna B. Karlsson1,6, Sergio Henrique Faria7,8,9, Vasileios Gkinis3, Helle Kjær3, Tobias Erhardt10, Sarah M. P. Berben11, Kerim H. Nisancioglu11,12, Iben Koldtoft3, and Sune Olander Rasmussen3

[revised manuscript text omitted]